# Error-independent effect of sensory uncertainty on motor learning when both feedforward and feedback control processes are engaged

Christopher L. Hewitson[1], David M. Kaplan [2,3], Matthew J. Crossley [2,3] *

**1** Department of Psychology, Yale University, New Haven, United States of America, **2** School of Psychological Sciences, Macquarie University, Sydney, Australia, **3** Macquarie University Performance and Expertise Research Centre, Macquarie University, Sydney, Australia

☯ These authors contributed equally to this work.
* matthew.crossley@mq.edu.au

**Data Availability Statement:** Data and analysis code can be accessed at: https://github.com/crossley/sensory_uncertainty_fffb.

## Abstract

Integrating sensory information during movement and adapting motor plans over successive movements are both essential for accurate, flexible motor behaviour. When an ongoing movement is off target, feedback control mechanisms update the descending motor commands to counter the sensed error. Over longer timescales, errors induce adaptation in feedforward planning so that future movements become more accurate and require less online adjustment from feedback control processes. Both the degree to which sensory feedback is integrated into an ongoing movement and the degree to which movement errors drive adaptive changes in feedforward motor plans have been shown to scale inversely with sensory uncertainty. However, since these processes have only been studied in isolation from one another, little is known about how they are influenced by sensory uncertainty in real-world movement contexts where they co-occur. Here, we show that sensory uncertainty may impact feedforward adaptation of reaching movements differently when feedback integration is present versus when it is absent. In particular, participants gradually adjust their movements from trial-to-trial in a manner that is well characterised by a slow and consistent envelope of error reduction. Riding on top of this slow envelope, participants exhibit large and abrupt changes in their initial movement vectors that are strongly correlated with the degree of sensory uncertainty present on the previous trial. However, these abrupt changes are insensitive to the magnitude and direction of the sensed movement error. These results prompt important questions for current models of sensorimotor learning under uncertainty and open up new avenues for future exploration in the field.

## Author summary

A large body of literature shows that sensory uncertainty inversely scales the degree of error-driven corrections made to motor plans from one trial to the next. However, by

---

**Funding:** The author(s) received no specific funding for this work.

**Competing interests:** The authors have no competing interests.

limiting sensory feedback to the endpoint of movements, these studies prevent corrections from taking place during the movement. Here, we show that when such corrections are promoted, sensory uncertainty punctuates between-trial movement corrections with abrupt changes that closely track the degree of sensory uncertainty but are insensitive to the magnitude and direction of movement error. This result marks a significant departure from existing findings and opens up new paths for future exploration.

## Introduction

During episodes of sensorimotor control, sensory information about the current state of the body and environment are used to generate motor commands to achieve a desired goal. In an ideal world, this process would be implemented perfectly and result in error-free motor behaviour. In the real world, however, every stage of the sensorimotor control process is contaminated by noise [1] and uncertainty [2]. Despite this, humans achieve remarkably accurate and appropriate motor behaviour by harnessing two complementary processes for error correction. First, sensory feedback is rapidly integrated to adjust ongoing movements and compensate for sensed deviations from the planned movement [3, 4]. Second, over successive movements, feedforward motor plans, which map behavioural goals to the motor commands needed to accomplish those goals, are adapted in response to movement errors [5, 6]. Throughout this paper, we refer to the former as *feedback integration* and the latter as *feedforward adaptation*[7].

An important question that has attracted attention recently concerns how these error-correction processes are influenced by sensory uncertainty. To date, most studies either investigate feedback integration or feedforward adaptation, but not both. For example, in their pioneering study, Körding and Wolpert [8] had participants perform a variation on a standard visuomotor adaptation task and showed that visual feedback provided briefly at the midpoint of the reach drives movement corrections that are inversely proportional to the level of uncertainty in the sensory feedback. In other words, when uncertainty is high, sensory feedback information is integrated less to correct the ongoing reach (and reliance on prior knowledge increases) compared to when uncertainty is low. Several follow-up studies have made similar observations about the influence of sensory uncertainty on feedback integration [9, 10].

Studies investigating the feedforward adaptation component have similarly shown that adaptation rates inversely scale with sensory uncertainty such that increasing sensory uncertainty leads to smaller updates to the feedforward plan (slower adaptation) and vice versa [9, 11–13]. These studies use endpoint feedback only and in doing so prevent feedback integration, effectively isolating the feedforward component.

Importantly, because the majority of studies investigate these processes in isolation (but see section "Can paradigm differences explain our divergent results?" for a more nuanced discussion of Körding and Wolpert [8]), little is known about how sensory uncertainty influences feedback integration and feedforward adaptation when they co-occur—as they do in most natural movement contexts. For example, if highly uncertain sensory feedback leads to relatively small online corrections during a movement, does it also drive similar adaptive changes in feedforward motor plans? To our knowledge, no existing studies address this key question.

Here, we examine how sensory uncertainty influences feedforward adaptation and feedback integration when they co-occur. Our results indicate that (1) the presence of feedforward adaptation has little to no effect on how sensory uncertainty influences feedback integration, but (2) in the presence of feedback integration, sensory uncertainty appears to punctuate a slow

and steady envelope of error reduction with large and abrupt changes to initial movement vectors that are insensitive to the magnitude and direction of the sensed movement error. This latter finding represents a significant departure from the existing literature, which consistently reports that sensory uncertainty inversely scales an error-dependent response.

## Results

The overarching aim of our experiments was to determine how different levels of sensory uncertainty impact feedforward adaptation and feedback integration when they co-occur. Participants made planar reaching movements using visual feedback about their hand position provided immediately before movement onset and at midpoint (Experiment 1), or immediately before movement onset, at midpoint, and at endpoint (Experiment 2 and 3). See Figs 1 and 2 for details. We quantify our behavioural results using traditional statistical methods (see the "Statistical modelling" section for details) as well as by fitting state-space models that make explicit assumptions about how motor commands are planned, executed, and updated over time, as well as how these processes are modulated by sensory uncertainty (see section "State-space modelling" for details.)

### Experiment 1

**Feedforward adaptation.**    Existing studies show that feedforward adaptation—when studied in the absence of feedback control—inversely scales with the level of uncertainty present in the sensory feedback [9, 11, 12, 14–17]. The primary contribution of Experiment 1 is to reveal how feedforward adaptation is influenced by sensory uncertainty when feedback integration is also engaged. To do this, we provided uncertain sensory feedback at movement midpoint (see Fig 2a), and omitted endpoint feedback entirely. We focused on uncertainty at midpoint

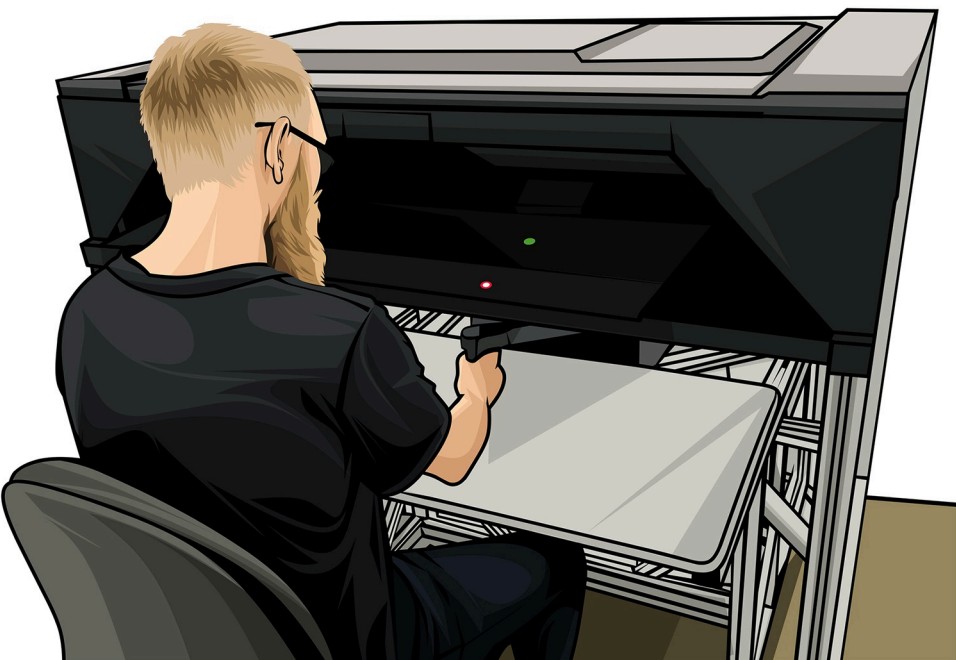

**Fig 1. Experimental apparatus and task structure.** Each participant made planar reaching movements while grasping the KINARM handle. A mirror system occluded vision of the hand and created the impression that the hand and visual targets were in the same plane. Red start target, green reach target, and white cursor feedback are shown.

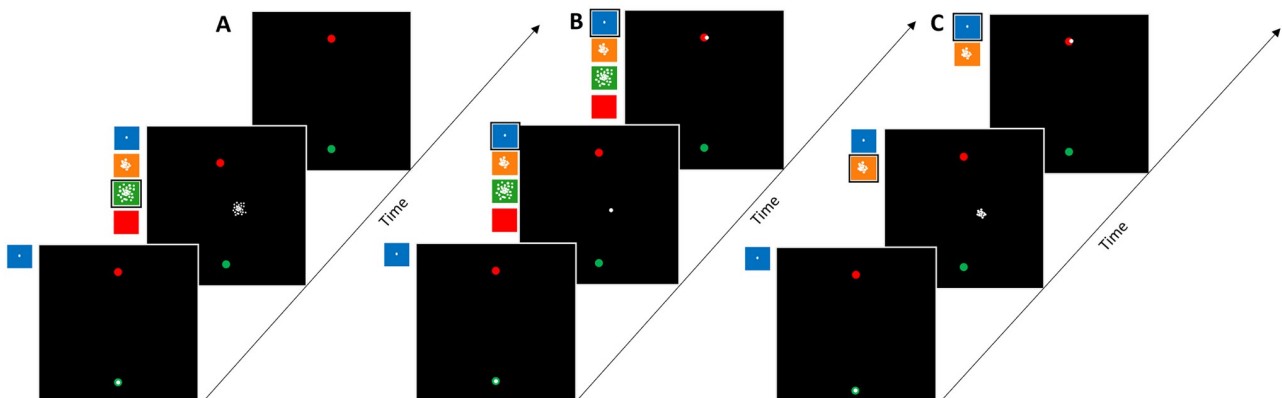

**Fig 2. Experimental protocols: Example adaptation phase trial conditions.** (A) Experiment 1: Midpoint-only feedback (large uncertainty). (B) Experiment 2: Matched midpoint and endpoint feedback (low uncertainty). (C) Experiment 3: Unmatched midpoint (moderate uncertainty) and endpoint (low uncertainty) feedback. Bottom, middle and top slides represent start, middle and end of reach respectively. coloured panels represent the possible uncertainty conditions (blue: $\sigma_L$, orange: $\sigma_M$, green: $\sigma_H$, red: $\sigma_\infty$). The example condition applied is outlined in black. In all experiments, a no-feedback washout phase followed the adaptation phase.

because feedback integration can only occur if feedback is provided at some point before movement offset. Presenting feedback briefly at midpoint is the simplest possible design in which feedback integration and feedforward adaptation both co-occur.

Previous computational and experimental work suggests that when the motor system performs feedback control, the feedback controller itself can be used as a teaching signal to drive adaptation in the feedforward controller [18–23]. It is therefore possible that feedback integration of sensory uncertainty at midpoint prevents or otherwise alters the influence of sensory uncertainty on feedforward adaptation. To our knowledge, this prediction has never been directly tested. Consequently, another key contribution of this experiment is to provide a clear test of this common assumption of computational models of motor learning.

Fig 3 shows group-averaged initial movement vectors for Experiment 1. Panel A is colour coded such that the colour of the dot at trial $t$ indicates the sensory uncertainty experienced at midpoint on trial $t-1$. Recall that the trial sequence of perturbations and sensory uncertainty levels were matched across participants, which is a fundamental feature of our experimental

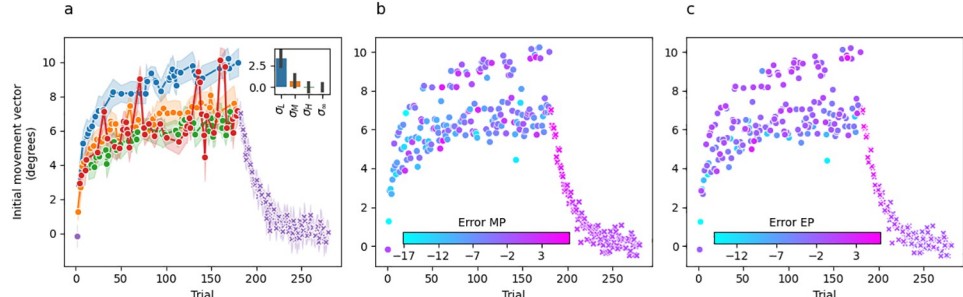

**Fig 3. Experiment 1 mean initial movement vector across participants per trial.** (a) Dot colour for trial $t$ represents the level of sensory uncertainty applied at midpoint on the previous trial $t-1$. Performance during the washout phase is shown by purple x's. The inset bar graph shows the mean difference between the last 10 trials of adaptation and the first 10 trials of washout plotted separately for each uncertainty level. Error bars are 95% confidence intervals. (b) Dot colour for trial $t$ represents the error at reach midpoint on the same trial. (c) Dot colour for trial $t$ represents the error at the reach endpoint on the previous trial $t-1$.

design that makes this plot informative. Washout trials are shown in purple, but from the participant's perspective they are identical to the unlimited uncertainty trials shown in red.

There are several key takeaways from Fig 3a. First, the change in initial movement vectors across trials during the adaptation phase is in a direction that—on average—tends to reduce error towards an adaptation extent of 7.13±1.9° (59%), averaged over the last 10 trials. Second, initial movement vectors decay smoothly back to baseline during the washout phase. Together, these observations are consistent with the idea that changes in initial movement vector across trials are driven by an incremental error-driven adaptive process. Third, and perhaps most strikingly, there is a clear stratification across sensory uncertainty conditions indicating that sensory uncertainty at midpoint has a dramatic and systematic effect on the evolution of initial movement vectors across trials.

Recall that the perturbation experienced on trial $t$ is on average 12° and therefore the error experienced on every trial should drive the subsequent trial's initial movement vector in a more positive direction. Instead, we see changes in movement vectors across trials from a lower uncertainty level to a higher uncertainty level (e.g., $\sigma_L \rightarrow \sigma_M$, $\sigma_M \rightarrow \sigma_H$) that are in a direction leading to greater error on the subsequent trial. Considered the other way around, any change across trials from a higher to a lower uncertainty level (e.g., $\sigma_M \rightarrow \sigma_L$, $\sigma_H \rightarrow \sigma_M$) almost always results in a change in initial movement vector that is adaptive (i.e., in an error-reducing direction), but of a much greater magnitude than would be seen if the uncertainty level did not change at all. Motor noise is not a plausible explanation for this pattern because the correlation between the level of uncertainty on the previous trial and movement direction on the current trial is consistent both across the experiment for individual participants and also across all participants.

According to either of the above perspectives, the stratification in initial movement vectors based on the previous trial's level of sensory uncertainty is difficult to reconcile with what is known about the incremental, error-driven nature of the implicit motor adaptation system. This naturally raises the question of whether the observed stratification reflects the process of motor adaptation at all, or may instead reflect the operation of some other system (e.g., whatever system is responsible for explicit aiming strategies [24]). Even though our paradigm was not designed to address this question, some insight can be gleaned by examining the no-feedback washout phase (shown in purple in Fig 3). For example, if initial movement vectors on low uncertainty trials (blue dots in Fig 3a) are an artifact of explicit aiming, there should be a large difference between the initial movement vectors observed for these trial types at the end of adaptation and those observed at the beginning of washout. Visual inspection of Fig 3 shows that this is indeed the case. To formalize this finding, we computed the difference between the mean accuracy achieved on the last 10 trials of the adaptation phase and the first 3 trials of the washout phase separately for each uncertainty trial type and separately for each subject. We used 3 washout trials instead of 10 in order to limit contamination from forgetting (i.e., the decay back to baseline seen during washout) which can lead to an underestimation of the initial washout state. A repeated-measures ANOVA indicated a significant difference in these difference scores across uncertainty trial types ($F(3, 57) = 27.64, p < .001, \eta_G^2 = 0.44$). Posthoc paired t-tests corrected for multiple comparisons using the Bonferroni method (see Table 1) revealed that these difference scores were larger for the low uncertainty condition than they were for any other uncertainty condition. These difference scores were not significantly different between any of the other uncertainty trial conditions.

Thus, the likely state of adaptation at the beginning of washout appears much more closely aligned with the adaptation estimated by initial movement vectors preceded by medium, high, and infinite uncertainty trials (orange, green, and red dots in Fig 3) than it does with the

**Table 1. Experiment 1 pairwise comparisons examining differences between uncertainty trial types in adaptation —washout difference scores.** *A* and *B* indicate the uncertainty trial types being compared; *T* is the observed t-statistic; *dof* is the degrees of freedom of the test; *p-corr* is the Bonferroni-corrected p-value; *hedges* is the Hedges G measure of effect size.

| row | A | B | T | dof | p-corr | hedges |
|---|---|---|---|---|---|---|
| 1 | $\sigma_L$ | $\sigma_M$ | 6.98 | 19.00 | 0.00 | 1.95 |
| 2 | $\sigma_L$ | $\sigma_H$ | 4.59 | 19.00 | 0.00 | 1.30 |
| 3 | $\sigma_L$ | $\sigma_\infty$ | 7.01 | 19.00 | 0.00 | 1.86 |
| 4 | $\sigma_M$ | $\sigma_H$ | -2.18 | 19.00 | 0.25 | -0.53 |
| 5 | $\sigma_M$ | $\sigma_{\text{infty}}$ | 0.16 | 19.00 | 1.00 | 0.04 |
| 6 | $\sigma_H$ | $\sigma_{\text{infty}}$ | 2.25 | 19.00 | 0.22 | 0.51 |

adaptation estimated by initial movement vectors preceded by low uncertainty trials (blue dots in Fig 3). This is consistent with the possibility that the blue dots in Fig 3a are the output of a process distinct from motor adaptation (see the "Adaptation vs aiming" subsection of the Discussion).

One small, apparent puzzle in our data is that several no-feedback trials ($\sigma_\infty$, in red) result in initial movement vectors on the subsequent trial that appear closely aligned with the low uncertainty trials ($\sigma_L$, in blue). These occur at trials 31, 71, 136, 137, 160 and 165 (Fig 3a). While this pattern could be due to noise, it is striking that out of these 6 trials, 5 are directly preceded by a low-uncertainty trial, and one (trial 137) is preceded by a no-feedback trial. Throughout adaptation, there are only 6 trial pairs where $\sigma_L$ precedes $\sigma_\infty$ (trials [[3,4], [30,31], [70,71], [135,136], [159,160], [164,165]]). Thus, in all but the earliest case (trials [3, 4]) a no-feedback trial following a low-uncertainty trial had the same effect as a low-uncertainty trial on subsequent initial movement vectors, suggesting that no-feedback trials may simply preserve the behaviour from the previous trial.

Fig 3b colour codes initial movement vector by the error experienced at midpoint on the previous trial, and Fig 3c colour codes by the error experienced at endpoint. Since no visual feedback was provided at endpoint in this experiment, participants could only estimate endpoint error based on proprioceptive feedback. Interestingly, the stratification pattern observed when colour coding by sensory uncertainty is no longer evident in either of these panels. Overall, this suggests that sensory uncertainty, and not movement error, is responsible for the stratification of initial movement vector across trials.

We formalised these observations by fitting a regression model that treated initial movement vector as the observed variable. Predictor variables in this model were trial, the error experienced at midpoint/endpoint, and the sensory uncertainty experienced at midpoint/endpoint, and the interaction between the error terms and the sensory uncertainty terms. All error and sensory uncertainty predictors were taken from the *previous* trial. See the "Statistical modelling" section for more details.

The predicted initial movement vectors from this regression model are shown in Fig 4a, with the best fitting beta coefficients along with their 95% confidence intervals shown by blue lines in Fig 4b. Table 2 includes all estimated beta coefficients and corresponds to the blue confidence intervals displayed in Fig 4b). The model was statistically significant (Adjusted $R^2 = 0.831$, $F_{(8,168)} = 103.1$, $p < .001$).

Sensory uncertainty across all levels significantly predicted the initial movement vector on the following trial in a direction that inversely scaled with uncertainty (Table 2, rows 2–4). Initial movement vectors were significantly greater on trials following low sensory uncertainty than they were following moderate sensory uncertainty ($\sigma_M - \sigma_L$; Table 2, row 2) and they

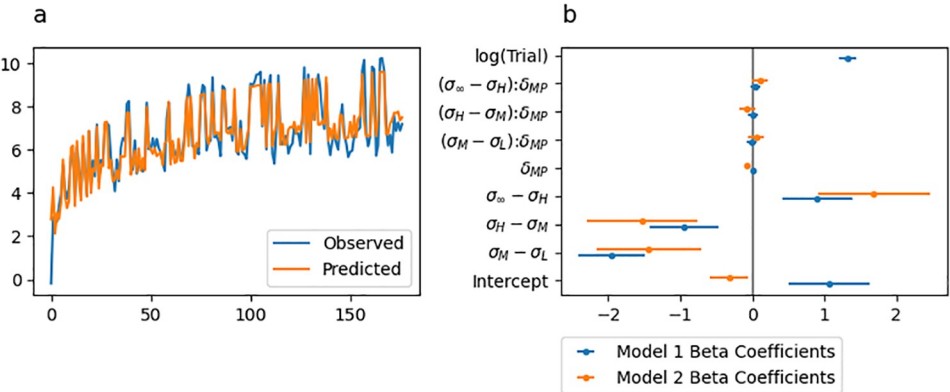

**Fig 4. Experiment 1 linear regression fit to initial movement vector.** (a) Initial movement vector predictions from the regression model superimposed over the behavioural data. (b) Point and 95% confidence interval estimates from best fitting regression models. Coefficients of the regression for predicting initial movement vector are shown in blue and coefficients for predicting change in initial movement vector are shown in orange.

were significantly greater on trials following moderate sensory uncertainty than they were following high sensory uncertainty ($\sigma_H - \sigma_M$; Table 2, row 3). However, they were significantly lower on trials following high sensory uncertainty than they were following unlimited sensory uncertainty ($\sigma_\infty - \sigma_H$; Table 2, row 4) possibly suggesting that the unlimited uncertainty condition should be treated in a qualitatively distinct fashion relative to the other uncertainty conditions. Error at midpoint ($\epsilon_{MP}$; Table 2, row 5) was not a significant predictor of initial movement vector (but see the results from the regression below examining *change* in initial movement vector). Furthermore, no interaction term between midpoint error and sensory uncertainty was significant (Table 2, rows 6–8). Finally, $log(Trial)$ significantly predicted initial movement vectors (Table 2, row 9), indicating that initial movement vectors tend to increase over the course of the adaptation phase.

The relative importance of the $log(Trial)$ term in the regression model was 0.47, and the relative importance of the sum of uncertainty terms was 0.25. This indicates that the slow envelope of the adaptation curve is well captured by a simple logarithmic function, but that the effect of sensory uncertainty captures substantial variance. Overall, this regression analysis revealed a clear effect of sensory uncertainty that is independent of the error experienced at

**Table 2. Experiment 1 regression results for predicting initial movement vector from error and sensory uncertainty terms.** These results correspond to the blue confidence intervals displayed in Fig 3. The *coef* column contains $\beta$ coefficients, the *se* column contains standard errors of these coefficients, the *T* column contains corresponding t-statistic, the *pval* column contains corresponding p-values, the *CI[2.5%]* and *CI[97.5%]* columns give the 95% confidence interval, and the *relimp* column gives the corresponding relative importance.

| row | names | coef | se | T | pval | CI[2.5%] | CI[97.5%] | relimp |
|---|---|---|---|---|---|---|---|---|
| 1 | $\beta_0$ | 1.06 | 0.29 | 3.69 | 0.00 | 0.50 | 1.63 | NaN |
| 2 | $\sigma_M - \sigma_L$ | -1.95 | 0.23 | -8.52 | 0.00 | -2.41 | -1.50 | 0.17 |
| 3 | $\sigma_H - \sigma_M$ | -0.95 | 0.24 | -3.93 | 0.00 | -1.43 | -0.47 | 0.07 |
| 4 | $\sigma_\infty - \sigma_H$ | 0.90 | 0.25 | 3.65 | 0.00 | 0.41 | 1.39 | 0.01 |
| 5 | $\delta_{MP}$ | 0.00 | 0.01 | 0.31 | 0.76 | -0.02 | 0.03 | 0.03 |
| 6 | $(\sigma_M - \sigma_L){:}\delta_{MP}$ | -0.01 | 0.04 | -0.28 | 0.78 | -0.08 | 0.06 | 0.05 |
| 7 | $(\sigma_H - \sigma_M){:}\delta_{MP}$ | 0.01 | 0.04 | 0.22 | 0.83 | -0.06 | 0.08 | 0.02 |
| 8 | $(\sigma_\infty - \sigma_H){:}\delta_{MP}$ | 0.04 | 0.04 | 1.16 | 0.25 | -0.03 | 0.12 | 0.01 |
| 9 | log(Trial) | 1.33 | 0.06 | 21.68 | 0.00 | 1.20 | 1.45 | 0.47 |

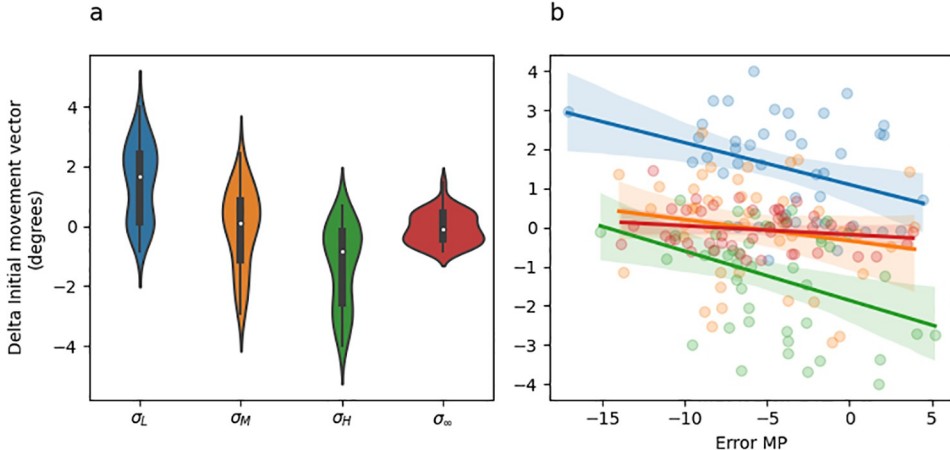

**Fig 5. Experiment 1 change in initial movement vector and linear regression fits.** (a) Violin plot depict in the distribution of mean changes in initial movement vector across all adaptation phase trials of the experiment separately for each uncertainty level. The inset of each violin shows a box plot in which the white dot indicates the median data value, the black box spans the 25% to 75% percentiles, and the whiskers extend to the most extreme data points. (b) Scatter plot showing the mean change in initial movement vector as a function of error experienced at midpoint. Lines indicate fitted simple linear regression lines. These regression lines do not correspond to the coefficients included in Table 3, rows 6–8) and are included only as a visual aid. colours indicate uncertainty level.

midpoint. We found no evidence that sensory uncertainty scales the response to movement error.

We also fit a regression model treating *change* in initial movement vector from trial to trial as the observed variable. Predictor variables for this model were identical to those described above, but with no predictor for trial. The best fitting beta coefficients along with their 95% confidence intervals are indicated by orange lines in Fig 4b. The model was statistically significant (Adjusted $R^2$ = 0.445, $F_{(7,169)}$ = 19.33, $p < .001$).

An intuition for what this analysis should reveal emerges from careful inspection of Fig 5a, which depicts change in initial movement colour coded by the sensory uncertainty at midpoint on the previous trial. Here it is evident that change in initial movement vectors inversely tracks sensory uncertainty levels with the exception of the no feedback trials ($\sigma_\infty$). This result can be seen in the regression results by noting that change in initial movement vectors were significantly greater on trials following low sensory uncertainty than they were following moderate sensory uncertainty ($\sigma_M - \sigma_L$; Table 3, row 2), they were significantly greater on trials following

**Table 3. Experiment 1 regression results for predicting change in initial movement vector from error and sensory uncertainty terms.** These results correspond to the orange confidence intervals displayed in Fig 3. The *coef* column contains $\beta$ coefficients, the *se* column contains standard errors of these coefficients, the *T* column contains corresponding t-statistic, the *pval* column contains corresponding p-values, the *CI[2.5%]* and *CI[97.5%]* columns give the 95% confidence interval, and the *relimp* column gives the corresponding relative importance.

| row | names | coef | se | T | pval | CI[2.5%] | CI[97.5%] | relimp |
|---|---|---|---|---|---|---|---|---|
| 1 | $\beta_0$ | -0.32 | 0.13 | -2.39 | 0.02 | -0.59 | -0.06 | NaN |
| 2 | $\sigma_M - \sigma_L$ | -1.44 | 0.37 | -3.92 | 0.00 | -2.16 | -0.71 | 0.17 |
| 3 | $\sigma_H - \sigma_M$ | -1.52 | 0.39 | -3.93 | 0.00 | -2.29 | -0.76 | 0.12 |
| 4 | $\sigma_\infty - \sigma_H$ | 1.68 | 0.39 | 4.27 | 0.00 | 0.90 | 2.46 | 0.04 |
| 5 | $\delta_{MP}$ | -0.08 | 0.02 | -3.80 | 0.00 | -0.12 | -0.04 | 0.04 |
| 6 | $(\sigma_M - \sigma_L){:}\delta_{MP}$ | 0.05 | 0.06 | 0.93 | 0.35 | -0.06 | 0.17 | 0.07 |
| 7 | $(\sigma_H - \sigma_M){:}\delta_{MP}$ | -0.07 | 0.06 | -1.26 | 0.21 | -0.19 | 0.04 | 0.03 |
| 8 | $(\sigma_\infty - \sigma_H){:}\delta_{MP}$ | 0.10 | 0.06 | 1.77 | 0.08 | -0.01 | 0.22 | 0.01 |

moderate sensory uncertainty than they were following high sensory uncertainty ($\sigma_H - \sigma_M$; Table 3, row 3), and they were significantly less on trials following high sensory uncertainty than they were following total sensory uncertainty ($\sigma_\infty - \sigma_H$; Table 3, row 4).

Further understanding of this effect can be built by examining Fig 5b, which depicts the change in initial movement vectors as a function of midpoint error. The lines in this plot represent simple linear regression lines (they are not the same multiple regression model described above) and are colour coded by sensory uncertainty. Note that if sensory uncertainty scales the response to error, the slope of the blue line (low uncertainty) should be the steepest, the slope of the orange line (moderate uncertainty) should be the next steepest, the slope of the green line (high uncertainty) should be the next steepest, and the slope of the red line should be the shallowest. Fig 5b does not show this pattern. See Table 3 rows 6–8 for statistics corresponding to the interaction between midpoint error and sensory uncertainty at midpoint. However, this analysis does reveal that midpoint error itself significantly contributes to predicting change in hand angle (Table 3 row 5). This can be seen in Fig 5b by noting that the overall trend of all regression lines is negative.

Overall, the results of this regression model (taking *change* in initial movement vector as the observed variable) largely echo the results of the regression model built with initial movement vector taken as the observed variable. Specifically, while midpoint error clearly influences the magnitude of hand angle change on the next trial, it does not interact with the level of sensory uncertainty. Furthermore, the sum of the relative importance of the sensory uncertainty predictors was much greater than that of midpoint error (0.33 vs 0.04). Thus, the main takeaway from these analyses is that both sensory uncertainty and movement error influence feedforward adaptation as estimated by initial movement vectors, but (1) they do so independently of each other, and (2) the effect of sensory uncertainty is more influential than the effect of movement error.

**Feedback integration.** Fig 6a shows feedback integration (i.e., the difference between midpoint and endpoint hand angles) as a function of sensory uncertainty at midpoint, and Fig 6b shows feedback integration as a function or error at midpoint coloured by sensory

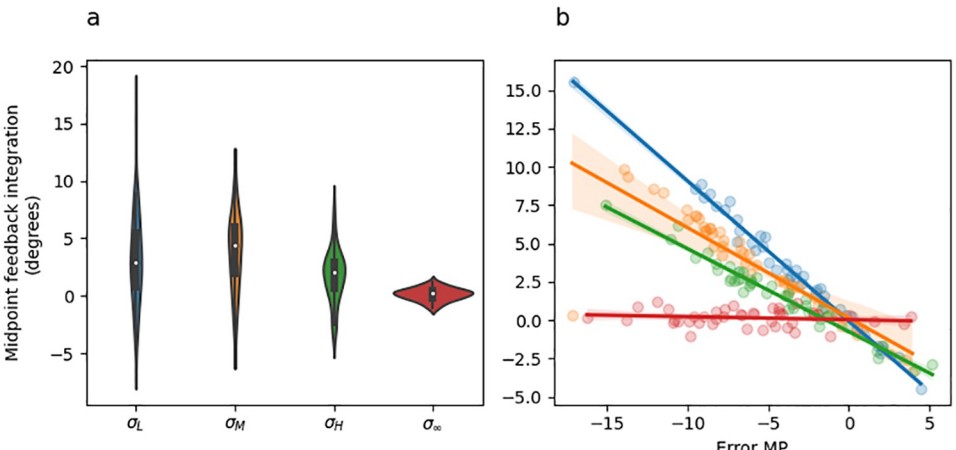

**Fig 6. Experiment 1 feedback integration (endpoint hand angle—initial movement vector).** (a) Violin plot depicting the distribution of mean feedback integration across all adaptation phase trials of the experiment separately for each uncertainty level. The inset of each violin shows a box plot in which the white dot indicates the median data value, the black box spans the 25% to 75% percentiles, and the whiskers extend to the most extreme data points. (b) Scatter plot showing the mean feedback integration as a function of error experienced at midpoint. Lines indicate fitted linear regression lines, corresponding to the coefficients included in Table 4, rows 6–8). Point and line colour indicates uncertainty level.

uncertainty. The main pattern observed in Fig 6 is that there are no significant differences in overall feedback integration depending on sensory uncertainty level (panel A), but there are large differences in how feedback integration responds to midpoint error depending on sensory uncertainty level (i.e., slope differences in panel B).

We formalised these observations by fitting a regression model treating the difference between endpoint and midpoint hand angle as the observed variable. Predictor variables were the error experienced at midpoint, the sensory uncertainty experienced at midpoint, and the interaction between these two terms. In contrast to the regression models reported for feedforward adaptation, all error and sensory uncertainty predictors were taken from the *current trial*. This regression was statistically significant (Adjusted $R^2 = 0.905$, $F_{(7,169)} = 241.5$, $p <$ .001). Beta coefficient estimates and corresponding statistics are listed in Table 4.

The most important result from this analysis is that the feedback integration response to increasing midpoint error (i.e., the slopes in Fig 6b) was significantly greater for low sensory uncertainty than it was for moderate uncertainty [$(\sigma_M - \sigma_L)$ * $\epsilon_{MP}$; Table 3 row 6], and was also significantly greater for high uncertainty than it was for unlimited uncertainty midpoint feedback [$(\sigma_M - \sigma_L)$ * $\epsilon_{MP}$; Table 3 row 8]. The difference between moderate and high uncertainty was non-significant [$(\sigma_H - \sigma_M)$ * $\epsilon_{MP}$; Table 3 row 7]. Overall, these results are consistent with prior studies showing that that sensory uncertainty has an error-scaling effect on feedback integration [8, 10].

## Experiment 2

The results of Experiment 1 raise the possibility that the uncertainty of sensory feedback does not inversely scale the magnitude of the error-driven feedforward update as it does in an endpoint feedback-only paradigm [11–13, 15–17]. Rather, our analysis suggests that sensory uncertainty acts largely independently of the experienced error to induce large and abrupt changes in the adapted state that can be—in the case of a transition from a high uncertainty trial—in a direction opposite to the general adaptive trend.

These striking results suggest that the presence of feedback integration fundamentally alters how feedforward adaptation is affected by sensory uncertainty. However, another possibility is that feedforward adaptation is influenced by sensory uncertainty differently depending on the temporal proximity of the sensory feedback signal to movement offset, regardless of whether or not feedback integration occurred at the midpoint of the movement [25–27]. Experiment 2 tests this possibility by providing midpoint and endpoint feedback matched in their level of uncertainty (see Fig 2B).

**Table 4. Experiment 1 regression results for predicting feedback integration (endpoint hand angle—initial movement vector) from error and sensory uncertainty terms.** The *coef* column contains $\beta$ coefficients, the *se* column contains standard errors of these coefficients, the *T* column contains corresponding t-statistic, the *pval* column contains corresponding p-values, the *CI[2.5%]* and *CI[97.5%]* columns give the 95% confidence interval, and the *relimp* column gives the corresponding relative importance.

| row | names | coef | se | T | pval | CI[2.5%] | CI[97.5%] | relimp |
|---|---|---|---|---|---|---|---|---|
| 1 | $\beta_0$ | -0.15 | 0.11 | -1.35 | 0.18 | -0.37 | 0.07 | NaN |
| 2 | $\sigma_M - \sigma_L$ | 0.21 | 0.30 | 0.70 | 0.48 | -0.39 | 0.81 | 0.01 |
| 3 | $\sigma_H - \sigma_M$ | -0.90 | 0.32 | -2.81 | 0.01 | -1.53 | -0.27 | 0.05 |
| 4 | $\sigma_\infty - \sigma_H$ | 0.78 | 0.32 | 2.41 | 0.02 | 0.14 | 1.42 | 0.05 |
| 5 | $\delta_{MP}$ | -0.51 | 0.02 | -30.96 | 0.00 | -0.55 | -0.48 | 0.49 |
| 6 | $(\sigma_M - \sigma_L){:}\delta_{MP}$ | 0.33 | 0.05 | 7.03 | 0.00 | 0.23 | 0.42 | 0.05 |
| 7 | $(\sigma_H - \sigma_M){:}\delta_{MP}$ | 0.05 | 0.05 | 1.01 | 0.31 | -0.05 | 0.14 | 0.11 |
| 8 | $(\sigma_\infty - \sigma_H){:}\delta_{MP}$ | 0.52 | 0.05 | 10.90 | 0.00 | 0.43 | 0.61 | 0.14 |

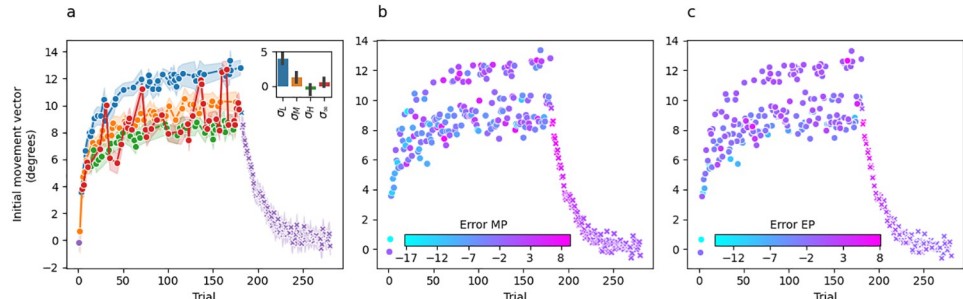

**Fig 7. Experiment 2 mean initial movement vector across participants per trial.** (a) The colour of the dot on trial $t$ represents the level of sensory uncertainty applied at midpoint on the previous trial $t − 1$. Performance during the washout phase is shown by purple x's. The inset bar graph shows the mean difference between the last 10 trials of adaptation and the first 10 trials of washout plotted separately for each uncertainty level. Error bars are 95% confidence intervals. (b) The colour of the dots on trial $t$ represents the error at the reach midpoint on the same trial. (c) The colour of the dots on trial $t$ represents the error at the reach endpoint on the previous trial $t − 1$.

Fig 7 shows group-averaged initial movement vectors per trial, colour coded such that the colour of the point at trial $t$ indicates the sensory uncertainty experienced at midpoint on trial $t − 1$. The extent of adaptation was 9.69±2.18˚ (81%) averaged over the last 10 trials.

**Feedforward adaptation.**   In terms of sensory uncertainty scaling, the same basic pattern observed in Experiment 1 (i.e. Fig 3) is present here as well. Specifically, the trial-by-trial variation in initial movement vector is large, but cannot be attributed to noise because there is a clear stratification in how a particular uncertainty level influences the subsequent trial. Even with the inclusion of congruent endpoint feedback, sensory uncertainty on trial $t − 1$ continues to exert a powerful effect on the subsequent initial movement vector.

As was already discussed for Experiment 1, a natural question arises about whether this stratification reflects implicit motor adaptation or some other process such as explicit aiming strategies [24]). We therefore followed the same logic as before and performed the same analysis outlined above. In particular, we computed the difference between the mean accuracy achieved on the last 10 trials of the adaptation phase and the first 3 trials of the washout phase separately for each uncertainty trial type and separately for each subject. A repeated-measures ANOVA indicated that there was a significant difference in these difference scores across uncertainty trial types ($F(3, 57) = 56.02, p < .001, \eta_G^2 = 0.5$). Posthoc paired t-tests corrected for multiple comparisons using the Bonferroni method (see Table 5) revealed that these difference scores were larger for the low uncertainty condition than they were for any other uncertainty condition. Additionally, the difference scores for the high uncertainty trial type (green dots in Fig 7a) were significantly smaller than those for the infinite uncertainty trial type (red

**Table 5. Experiment 2 pairwise comparisons examining differences between uncertainty trial types in adaptation —washout difference scores.** $A$ and $B$ indicate the uncertainty trial types being compared; $T$ is the observed t-statistic; *dof* is the degrees of freedom of the test; *p-corr* is the Bonferroni-corrected p-value; *hedges* is the Hedges G measure of effect size.

| row | A | B | T | dof | p-corr | hedges |
|---|---|---|---|---|---|---|
| 1 | $\sigma_L$ | $\sigma_M$ | 10.61 | 19.00 | 0.00 | 2.06 |
| 2 | $\sigma_L$ | $\sigma_H$ | 7.11 | 19.00 | 0.00 | 1.56 |
| 3 | $\sigma_L$ | $\sigma_\infty$ | 10.53 | 19.00 | 0.00 | 2.49 |
| 4 | $\sigma_M$ | $\sigma_H$ | -2.43 | 19.00 | 0.15 | -0.42 |
| 5 | $\sigma_M$ | $\sigma_{\text{infty}}$ | 2.92 | 19.00 | 0.05 | 0.64 |
| 6 | $\sigma_H$ | $\sigma_{\text{infty}}$ | 4.77 | 19.00 | 0.00 | 0.99 |

dots in Fig 7a). These difference scores were not significantly different between any of the other uncertainty trial conditions.

As seen in Experiment 1, this is consistent with the idea that the likely state of adaptation at the beginning of washout is more closely aligned with the adaptation estimated by initial movement vectors preceded by medium, high, and unlimited uncertainty trials (orange, green, and red dots in Fig 7a) than it does with the adaptation estimated by initial movement vectors preceded by low uncertainty trials (blue dots in Fig 7a). In this case, adaptation appears most aligned with the level estimated by high uncertainty trials. These findings are consistent with the possibility that the blue dots in Fig 7a are the output of a process distinct from motor adaptation (see the "Adaptation vs aiming" subsection of the Discussion).

Fig 7b and 7c depict initial movement vector colour coded by the error experienced at midpoint or endpoint, respectively. As in Experiment 1, these panels show a clear pattern of stratification by sensory uncertainty and no stratification by movement error, suggesting that sensory uncertainty (and not movement error) is responsible for the stratification of initial movement vector observed across trials. As in Experiment 1, the no-feedback trials ($\sigma_\infty$, in red) that follow low-uncertainty trials ($\sigma_L$, in blue) have the same effect on subsequent initial movement vectors as a low uncertainty trial, again suggesting that no-feedback trials may preserve the behaviour from the previous trial.

We fit a regression model of the same form as that reported for Experiment 1 (see also the "Statistical modelling" section). The predicted initial movement vectors from this regression model is shown in Fig 8a and the best fitting beta coefficients along with their 95% confidence intervals are shown by blue lines in Fig 8b. Table 6 includes all estimated beta coefficients and corresponds to the blue confidence intervals displayed in Fig 8b). The model was statistically significant (Adjusted $R^2 = 0.845$, $F_{(12,165)} = 81.17$, $p < .001$).

As in Experiment 1, sensory uncertainty across all levels significantly predicted the initial movement vector on the following trial in a direction that inversely scaled with uncertainty (Table 6, rows 2–4). Initial movement vectors were significantly greater on trials following low sensory uncertainty than they were following moderate sensory uncertainty ($\sigma_M - \sigma_L$; Table 6, row 2) and they were significantly greater on trials following moderate sensory uncertainty than they were following high sensory uncertainty ($\sigma_H - \sigma_M$; Table 6, row 3). However, they were significantly smaller on trials following high sensory uncertainty than they were following total sensory uncertainty ($\sigma_\infty - \sigma_H$; Table 6, row 4), possibly suggesting that total uncertainty is playing a qualitatively distinct role here than that for the other uncertainty conditions.

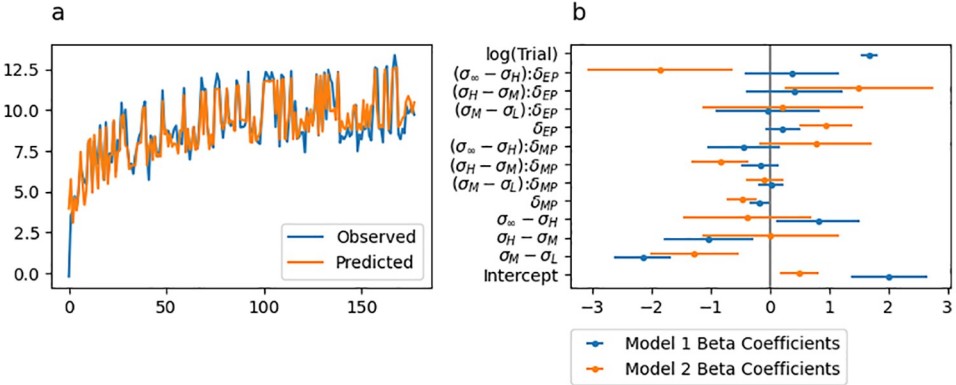

**Fig 8. Experiment 2 linear regression fit to initial movement vector.** (a) Initial movement vector predictions from the regression model superimposed over the behavioural data. (b) Point and 95% confidence interval estimates from best fitting regression models. Coefficients of the regression for predicting initial movement vector are shown in blue and coefficients for predicting change in initial movement vector are shown in orange.

**Table 6. Experiment 2 regression results for predicting initial movement vector from error and sensory uncertainty terms.** These results correspond to the blue confidence intervals displayed in Fig 8. The *coef* column contains $\beta$ coefficients, the *se* column contains standard errors of these coefficients, the *T* column contains corresponding t-statistic, the *pval* column contains corresponding p-values, the *CI[2.5%]* and *CI[97.5%]* columns give the 95% confidence interval, and the *relimp* column gives the corresponding relative importance.

| row | names | coef | se | T | pval | CI[2.5%] | CI[97.5%] | relimp |
|---|---|---|---|---|---|---|---|---|
| 1 | $\beta_0$ | 2.01 | 0.33 | 6.12 | 0.00 | 1.36 | 2.66 | NaN |
| 2 | $\sigma_M - \sigma_L$ | -2.15 | 0.25 | -8.76 | 0.00 | -2.64 | -1.67 | 0.14 |
| 3 | $\sigma_H - \sigma_M$ | -1.05 | 0.38 | -2.74 | 0.01 | -1.81 | -0.29 | 0.07 |
| 4 | $\sigma_\infty - \sigma_H$ | 0.81 | 0.36 | 2.25 | 0.03 | 0.10 | 1.52 | 0.01 |
| 5 | $\delta_{MP}$ | -0.17 | 0.08 | -2.10 | 0.04 | -0.34 | -0.01 | 0.03 |
| 6 | $(\sigma_M - \sigma_L){:}\delta_{MP}$ | 0.01 | 0.11 | 0.14 | 0.89 | -0.19 | 0.22 | 0.01 |
| 7 | $(\sigma_H - \sigma_M){:}\delta_{MP}$ | -0.17 | 0.16 | -1.05 | 0.29 | -0.48 | 0.15 | 0.01 |
| 8 | $(\sigma_\infty - \sigma_H){:}\delta_{MP}$ | -0.45 | 0.31 | -1.44 | 0.15 | -1.07 | 0.17 | 0.01 |
| 9 | $\delta_{EP}$ | 0.21 | 0.15 | 1.38 | 0.17 | -0.09 | 0.50 | 0.05 |
| 10 | $(\sigma_M - \sigma_L){:}\delta_{EP}$ | -0.04 | 0.45 | -0.09 | 0.92 | -0.92 | 0.84 | 0.03 |
| 11 | $(\sigma_H - \sigma_M){:}\delta_{EP}$ | 0.41 | 0.42 | 0.98 | 0.33 | -0.41 | 1.23 | 0.02 |
| 12 | $(\sigma_\infty - \sigma_H){:}\delta_{EP}$ | 0.37 | 0.40 | 0.91 | 0.37 | -0.43 | 1.16 | 0.01 |
| 13 | log(Trial) | 1.68 | 0.07 | 23.32 | 0.00 | 1.54 | 1.82 | 0.47 |

Error at midpoint ($\epsilon_{MP}$; Table 6, row 5), but not error at endpoint ($\epsilon_{MP}$; Table 6, row 9), was a significant predictor of initial movement vector. Importantly, no interaction terms between midpoint/endpoint error and sensory uncertainty (Table 6 rows 6–8 and 10–12) were significant. Finally, $log(Trial)$ significantly predicted initial movement vectors (6, row 13), indicating the trend of initial movement vectors to increase over the course of the adaptation phase. The relative importance of the $log(Trial)$ term was 0.47, that for the sum of uncertainty terms was 0.22, and that for the midpoint error ($\delta_{MP}$) term was 0.03. This suggests that the dominant source of variance is largely captured by the slow envelope of adaptation, but also shows that sensory uncertainty plays an important role.

Overall, this regression echoed the results of Experiment 1 in revealing (1) both error and sensory uncertainty influence adaptation, but (2) they exert their influences independently of each other. We therefore failed to find any evidence that sensory uncertainty scales the response to movement error.

We also fit a regression model treating *change* in initial movement vector as the observed variable. Predictor variables for this model were identical to those described above, but with no trial predictor. The best fitting betas along with their 95% confidence intervals is shown by the orange lines in Fig 4b. The model was statistically significant (Adjusted $R^2 = 0.522$, $F_{(11,166)} = 18.54$, $p < .001$).

A strong expectation for what this analysis should reveal can be built with careful inspection of Fig 9a, which depicts change in initial movement colour coded by the sensory uncertainty at midpoint on the previous trial. Here it is clear that change in initial movement vectors inversely tracks sensory uncertainty levels with the exception of the no feedback trials ($\sigma_\infty$). This result can be seen in the regression results by noting that change in initial movement vectors were significantly greater on trials following low sensory uncertainty than they were following moderate sensory uncertainty ($\sigma_M - \sigma_L$; Table 7, row 2). However, they were not significantly greater on trials following moderate sensory uncertainty than they were following high sensory uncertainty ($\sigma_H - \sigma_M$; Table 7, row 3), nor were they significantly greater on trials following high sensory uncertainty than they were following unlimited sensory uncertainty ($\sigma_\infty - \sigma_H$; Table 7, row 4). Thus, while the clear trend is for stratification according to sensory uncertainty, only the low uncertainty comparison reached significance.

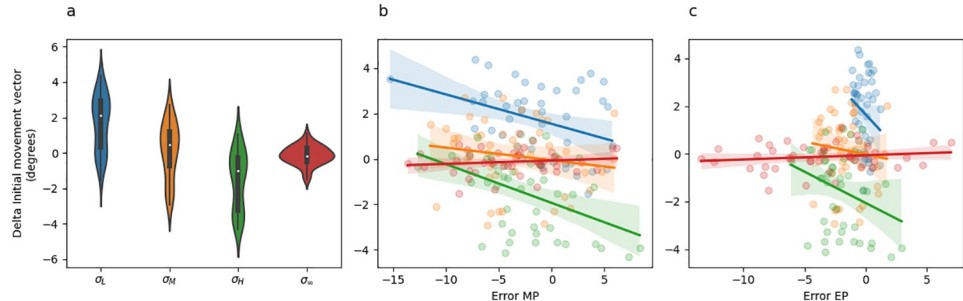

**Fig 9. Experiment 2 change in initial movement vector and linear regression fits.** (a) Violin plot depicting the distribution of mean changes in initial movement vector across all adaptation phase trials of the experiment separately for each midpoint/endpoint uncertainty combination, colour coded as per Fig 6a. The inset of each violin shows a box plot in which the white dot indicates the median data value, the black box spans the 25% to 75% percentiles, and the whiskers extend to the most extreme data points. (b) Scatter plot showing the mean change in initial movement vector as a function of error experienced at midpoint. Point and line colour indicates uncertainty level. (c) Scatter plot showing the mean change in initial movement vector as a function of error experienced at endpoint on the previous trial. Point and line colour indicates uncertainty level. The lines in panel B and C indicate fitted simple linear regression lines. These regression lines do not correspond to the coefficients included in Table 7) and are included only as a visual aid.

Further intuitions can be built by examining Fig 9b, which depicts the change in initial movement vectors as a function of midpoint error. The lines in this plot are colour coded by sensory uncertainty. Fig 9c shows essentially the same information but for endpoint instead of midpoint error. Note that if sensory uncertainty scales the response to error, then the slope of the blue line (low uncertainty) should be the steepest, the slope of the orange line (moderate uncertainty) should be the next steepest, the slope of the green line (high uncertainty) should be the next steepest, and the slope of the red line should be the shallowest. Neither Fig 5b nor Fig 9c show this pattern. See Table 7 rows 6–8 for statistics corresponding to midpoint error, and rows 10–12 for statistics corresponding to endpoint error.

Overall, the results of this regression (using *change* in initial movement vector as the observed variable) echo the results of the regression modelling using initial movement vector as the observed variable. Furthermore, the results from both regression models are aligned with those from Experiment 1. Specifically, there appears to be a clear error-independent effect

**Table 7. Experiment 2 regression results for predicting change in initial movement vector from error and sensory uncertainty terms.** These results correspond to the orange confidence intervals displayed in Fig 8. The *coef* column contains $\beta$ coefficients, the *se* column contains standard errors of these coefficients, the *T* column contains corresponding t-statistic, the *pval* column contains corresponding p-values, the *CI[2.5%]* and *CI[97.5%]* columns give the 95% confidence interval, and the *relimp* column gives the corresponding relative importance.

| row | names | coef | se | T | pval | CI[2.5%] | CI[97.5%] | relimp |
|---|---|---|---|---|---|---|---|---|
| 1 | $\beta_0$ | 0.49 | 0.17 | 2.95 | 0.00 | 0.16 | 0.82 | NaN |
| 2 | $\sigma_M - \sigma_L$ | -1.28 | 0.38 | -3.40 | 0.00 | -2.02 | -0.54 | 0.14 |
| 3 | $\sigma_H - \sigma_M$ | 0.01 | 0.59 | 0.02 | 0.98 | -1.15 | 1.17 | 0.11 |
| 4 | $\sigma_\infty - \sigma_H$ | -0.38 | 0.55 | -0.69 | 0.49 | -1.47 | 0.71 | 0.02 |
| 5 | $\delta_{MP}$ | -0.48 | 0.13 | -3.76 | 0.00 | -0.73 | -0.23 | 0.06 |
| 6 | $(\sigma_M - \sigma_L){:}\delta_{MP}$ | -0.09 | 0.16 | -0.58 | 0.57 | -0.41 | 0.23 | 0.02 |
| 7 | $(\sigma_H - \sigma_M){:}\delta_{MP}$ | -0.84 | 0.24 | -3.44 | 0.00 | -1.32 | -0.36 | 0.03 |
| 8 | $(\sigma_\infty - \sigma_H){:}\delta_{MP}$ | 0.77 | 0.48 | 1.60 | 0.11 | -0.18 | 1.72 | 0.03 |
| 9 | $\delta_{EP}$ | 0.94 | 0.23 | 4.06 | 0.00 | 0.48 | 1.40 | 0.04 |
| 10 | $(\sigma_M - \sigma_L){:}\delta_{EP}$ | 0.21 | 0.69 | 0.31 | 0.76 | -1.15 | 1.57 | 0.02 |
| 11 | $(\sigma_H - \sigma_M){:}\delta_{EP}$ | 1.50 | 0.64 | 2.35 | 0.02 | 0.24 | 2.76 | 0.02 |
| 12 | $(\sigma_\infty - \sigma_H){:}\delta_{EP}$ | -1.85 | 0.62 | -2.99 | 0.00 | -3.07 | -0.63 | 0.04 |

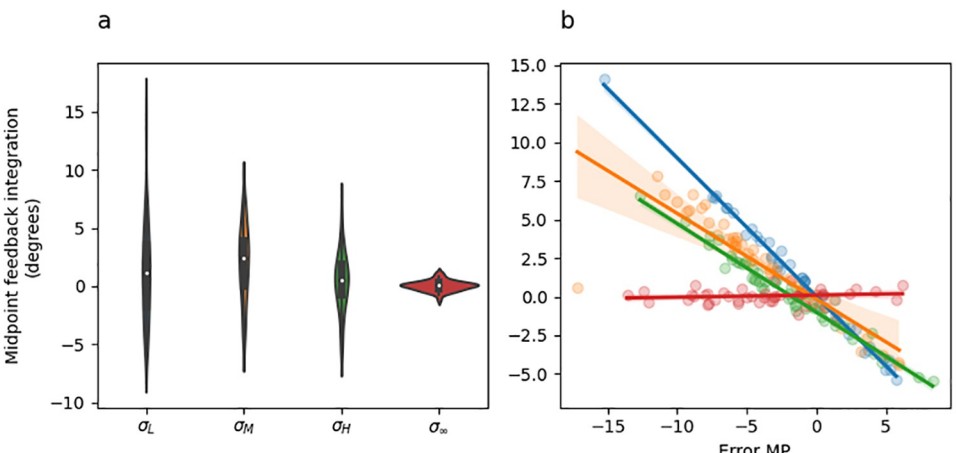

**Fig 10. Experiment 2 feedback integration (endpoint hand angle—initial movement vector).** (a) Violin plot depicting the distribution of mean feedback integration across all adaptation phase trials of the experiment separately for each uncertainty level. The inset of each violin shows a box plot in which the white dot indicates the median data value, the black box spans the 25% to 75% percentiles, and the whiskers extend to the most extreme data points. (b) Scatter plot showing the mean feedback integration as a function of error experienced at midpoint. Lines indicate fitted linear regression lines, corresponding to the coefficients included in Table 8, rows 6–8). Point and line colour indicates uncertainty level.

of sensory uncertainty—albeit less pronounced statistically then in the results from Experiment 1. Importantly, we find no evidence that sensory uncertainty scales the response to movement error.

**Feedback integration.** Fig 10a shows feedback integration (i.e., the difference between endpoint and midpoint hand angle) as a function of sensory uncertainty at midpoint, and Fig 10b shows feedback integration as a function or error at midpoint coloured by sensory uncertainty. The main pattern observed in Fig 10 is that there are no significant differences in overall feedback integration depending on sensory uncertainty level (panel A), but there are large differences in how feedback integration responds to midpoint error depending on sensory uncertainty level (i.e., slope differences in panel B).

We formalised these observations by fitting a regression model treating the difference between endpoint and midpoint hand angle as the observed variable. Predictor variables were the error experienced at midpoint, the sensory uncertainty experienced at midpoint, and the interaction between these two terms. In contrast to the regression models reported for feedforward adaptation, all error and sensory uncertainty predictors were taken from the *current trial*. This regression was statistically significant (Adjusted $R^2 = 0.911$, $F_{(7,170)} = 258.7$, $p < .001$). Beta coefficient estimates and corresponding statistics are listed in Table 8.

The most important result from this analysis is that the response of feedback integration to increasing midpoint error (i.e., the slopes in Fig 10b) was significantly greater for low sensory uncertainty than it was for moderate uncertainty [$(\sigma_M - \sigma_L) * \epsilon_{MP}$; Table 8 row 6], and was also significantly greater for high uncertainty than it was for totally uncertain midpoint feedback [$(\sigma_M - \sigma_L) * \epsilon_{MP}$; Table 8 row 8]. The difference between moderate and high uncertainty was non-significant [$(\sigma_H - \sigma_M) * \epsilon_{MP}$; Table 8 row 7].

Overall, the pattern of feedback integration seen in Experiment 2 are qualitatively identical to those observed in Experiment 1. Both are consistent with prior studies showing that that sensory uncertainty has an error-scaling effect on feedback integration [8, 10].

**Table 8. Experiment 2 regression results for predicting feedback integration (endpoint hand angle—initial movement vector) from error and sensory uncertainty terms.** The *coef* column contains $\beta$ coefficients, the *se* column contains standard errors of these coefficients, the *T* column contains corresponding t-statistic, the *pval* column contains corresponding p-values, the *CI[2.5%]* and *CI[97.5%]* columns give the 95% confidence interval, and the *relimp* column gives the corresponding relative importance.

| row | names | coef | se | T | pval | CI[2.5%] | CI[97.5%] | relimp |
|---|---|---|---|---|---|---|---|---|
| 1 | $\beta_0$ | -0.26 | 0.08 | -3.40 | 0.00 | -0.41 | -0.11 | NaN |
| 2 | $\sigma_M - \sigma_L$ | -0.16 | 0.21 | -0.74 | 0.46 | -0.58 | 0.26 | 0.01 |
| 3 | $\sigma_H - \sigma_M$ | -0.85 | 0.22 | -3.83 | 0.00 | -1.28 | -0.41 | 0.02 |
| 4 | $\sigma_\infty - \sigma_H$ | 1.11 | 0.22 | 5.02 | 0.00 | 0.67 | 1.55 | 0.02 |
| 5 | $\delta_{MP}$ | -0.50 | 0.01 | -34.78 | 0.00 | -0.53 | -0.47 | 0.60 |
| 6 | $(\sigma_M - \sigma_L){:}\delta_{MP}$ | 0.34 | 0.04 | 8.54 | 0.00 | 0.26 | 0.42 | 0.05 |
| 7 | $(\sigma_H - \sigma_M){:}\delta_{MP}$ | -0.02 | 0.04 | -0.45 | 0.65 | -0.10 | 0.06 | 0.08 |
| 8 | $(\sigma_\infty - \sigma_H){:}\delta_{MP}$ | 0.59 | 0.04 | 14.10 | 0.00 | 0.50 | 0.67 | 0.14 |

## Experiment 3

In Experiment 3, we sought to further probe how sensory uncertainty influences feedback and feedforward control processes. In particular, Experiment 3 disassociates the sensory uncertainty experienced at midpoint from that experienced at endpoint (see Fig 2c). This allows us to investigate if sensory uncertainty at midpoint dominates sensory uncertainty at endpoint (as might be expected due to the feedback correction made at that time point), or if endpoint dominates midpoint (as might be expected due to its temporal proximity to movement offset) [25, 27].

Fig 11a shows group-averaged initial movement vectors per trial colour coded such that the colour of the point at trial $t$ indicates the midpoint and endpoint sensory uncertainty combinations ($\sigma_{LL}$, $\sigma_{LH}$, $\sigma_{HL}$, $\sigma_{HH}$) experienced on trial $t - 1$. Mean adaptation extent over the last 10 trials was 11.91±2.29˚ (99%). There is a clear stratification between trial types according to endpoint uncertainty, regardless of midpoint uncertainty. Furthermore, similar to Experiment 1 and 2, we see that in the transition from lower endpoint uncertainty trials to higher endpoint uncertainty trials (e.g., $\sigma_{LL} \rightarrow \sigma_{LH}$; $\sigma_{HL} \rightarrow \sigma_{HH}$) the change in movement vector is in a direction that tends to increase error on the subsequent trial.

**Feedforward adaptation.** As noted in the Results section of Experiment 1 and 2, it is possible that the stratification of initial movement vector by endpoint hand angle reflects implicit

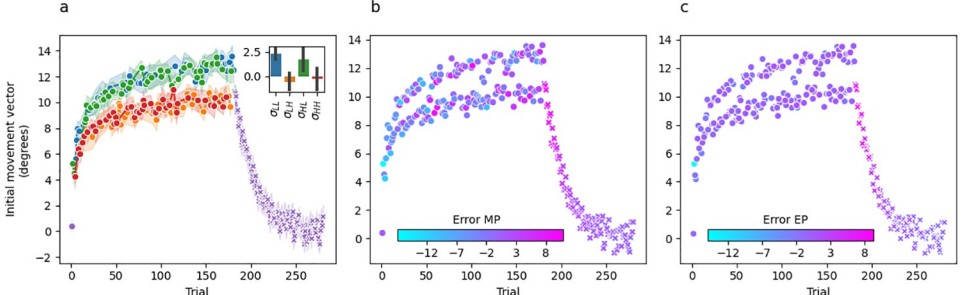

**Fig 11. Experiment 3 mean initial movement vector across participants per trial.** (a) The colour of the dot on trial $t$ represents the combination of sensory uncertainty applied at midpoint and endpoint on the previous trial $t - 1$. Specifically, $\sigma_{LL}$ in blue, $\sigma_{LH}$ in orange, $\sigma_{HL}$ in green, and $\sigma_{HH}$ in red. Performance during the washout phase is shown by purple x's. The inset bar graph shows the mean difference between the last 10 trials of adaptation and the first 10 trials of washout plotted separately for each trial type. Error bars are 95% confidence intervals. (b) The colour of the dots on trial $t$ represents the error at the reach midpoint on the previous trial $t - 1$. (c) The colour of the dots on trial $t$ represents the error at the reach endpoint on the previous trial $t - 1$.

**Table 9. Experiment 3 pairwise comparisons examining differences between uncertainty trial types in adaptation—washout difference scores.** *A* and *B* indicate the uncertainty trial types being compared; *T* is the observed t-statistic; *dof* is the degrees of freedom of the test; *p-corr* is the Bonferroni-corrected p-value; *hedges* is the Hedges G measure of effect size.

| row | A | B | T | dof | p-corr | hedges |
|---|---|---|---|---|---|---|
| 1 | $\sigma_L$ | $\sigma_M$ | 3.39 | 19.00 | 0.02 | 0.78 |
| 2 | $\sigma_L$ | $\sigma_H$ | -1.03 | 19.00 | 1.00 | -0.25 |
| 3 | $\sigma_L$ | $\sigma_\infty$ | 4.19 | 19.00 | 0.00 | 0.98 |
| 4 | $\sigma_M$ | $\sigma_H$ | -5.33 | 19.00 | 0.00 | -1.29 |
| 5 | $\sigma_M$ | $\sigma_{\mathrm{infty}}$ | 0.62 | 19.00 | 1.00 | 0.13 |
| 6 | $\sigma_H$ | $\sigma_{\mathrm{infty}}$ | 9.87 | 19.00 | 0.00 | 1.71 |

adaptation in the motor system. But it is also possible that it reflects the operation of some other system or process such as explicit aiming strategies [24]. We therefore followed the same logic and performed the same analysis outlined in earlier sections for those experiments. Visual inspection of Fig 11a reveals that initial movement vectors at the beginning of washout are closely matched to those observed at the end of adaptation for the trial types containing high uncertainty at endpoint. To formalise this observation, we computed the difference between the mean accuracy achieved on the last 10 trials of the adaptation phase and the first 3 trials of the washout phase separately for each uncertainty trial type and separately for each subject. A repeated-measures ANOVA indicated that there was a significant difference in these difference scores across uncertainty trial types ($F(3, 57) = 16.55, p < .001, \eta_G^2 = 0.26$). Posthoc paired t-tests corrected for multiple comparisons using the bonferoni method (see Table 9) revealed that these difference scores were larger for trials in high high uncertainty was provided at endpoint than for trials in which low uncertainty was provided at endpoint. The uncertainty provided at midpoint did not make difference.

These findings are consistent with the idea that the likely state of adaptation at the beginning of washout is more closely aligned with the adaptation estimated by initial movement vectors preceded by high endpoint uncertainty trials (orange and red dots in Fig 11a) than it does with the adaptation estimated by initial movement vectors preceded by low endpoint uncertainty trials (blue and green dots in Fig 11a). Thus, the adaptation envelope seen with low uncertainty trials may reflect the output of a process distinct from motor adaptation (see the "Adaptation vs aiming" subsection for further discussion).

Fig 11b and 11c depict initial movement vector colour coded by the error experienced at midpoint or endpoint, respectively. As in Experiment 1 and 2, these panels show that the stratification seen when colour coding by sensory uncertainty trial type is not present when colour coding by error, and suggests that sensory uncertainty and not movement error is responsible for the striation of initial movement vector across trials.

We formalised these observations by fitting a regression model of the same form as that reported in Experiment 1 and 2 (see also the "Statistical modelling" section). The predicted initial movement vectors from this regression model is shown in Fig 12a and the best fitting betas along with their 95% confidence intervals is shown by blue lines in Fig 12b. Table 10 includes all estimated beta coefficients and corresponds to the blue confidence intervals displayed in Fig 8b). The model was statistically significant (Adjusted $R^2 = 0.917$, $F_{(12,165)} = 163.2$, $p < .001$).

As in Experiment 1 and 2, sensory uncertainty across all conditions significantly influenced the initial movement vector on the following trial. Here, this influence was to increase initial movement vectors after low uncertainty at endpoint and to decrease them after high uncertainty at endpoint (Table 10, rows 2–4). Initial movement vectors were significantly greater on

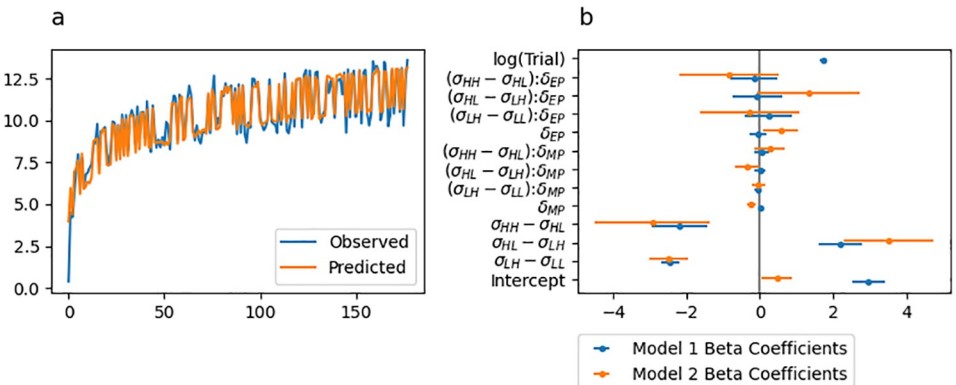

**Fig 12. Experiment 3 linear regression fit to initial movement vector.** (a) Initial movement vector predictions from the regression model superimposed over the behavioural data. (b) Point and 95% confidence interval estimates from best fitting regression models. Coefficients of the regression for predicting initial movement vector are shown in blue and coefficients for predicting change in initial movement vector are shown in orange.

trials following low endpoint sensory uncertainty than they were following high endpoint sensory uncertainty ($\sigma_{LH} - \sigma_{LL}$ and $\sigma_{HH} - \sigma_{HL}$; Table 10, rows 2 and 4). Neither error at midpoint ($\epsilon_{MP}$; Table 10, row 5) nor error at endpoint ($\epsilon_{EP}$; Table 10, row 9) were significant predictors of initial movement vector. Importantly, no interaction terms between midpoint/endpoint error and sensory uncertainty (Table 10 rows 6–8 and 10–12) were significant. Finally, *log* (*Trial*) significantly predicted initial movement vectors (Table 10, row 13), indicating the trend of initial movement vectors to increase over the course of the adaptation phase. As in Experiment 1 and 2 the relative importance of the *log*(*Trial*) term was 0.56 and the sum of the relative importance of the uncertainty terms was 0.24 indicating that both captured important variance in the data.

Overall, this regression echoed the results of Experiments 1 and 2, revealing that the effect of sensory uncertainty on feedforward adaptation is independent of movement error and failing to provide any evidence that sensory uncertainty scales the response to movement error.

**Table 10. Experiment 3 regression results for predicting initial movement vector from error and sensory uncertainty terms.** These results correspond to the blue confidence intervals displayed in Fig 12. The *coef* column contains $\beta$ coefficients, the *se* column contains standard errors of these coefficients, the *T* column contains corresponding t-statistic, the *pval* column contains corresponding p-values, the *CI[2.5%]* and *CI[97.5%]* columns give the 95% confidence interval, and the *relimp* column gives the corresponding relative importance.

| row | names | coef | se | T | pval | CI[2.5%] | CI[97.5%] | relimp |
|---|---|---|---|---|---|---|---|---|
| 1 | $\beta_0$ | 2.97 | 0.23 | 13.00 | 0.00 | 2.51 | 3.42 | NaN |
| 2 | $\sigma_{LH} - \sigma_{LL}$ | -2.45 | 0.13 | -19.39 | 0.00 | -2.70 | -2.20 | 0.16 |
| 3 | $\sigma_{HL} - \sigma_{LH}$ | 2.21 | 0.30 | 7.37 | 0.00 | 1.62 | 2.80 | 0.02 |
| 4 | $\sigma_{HH} - \sigma_{HL}$ | -2.20 | 0.38 | -5.79 | 0.00 | -2.95 | -1.45 | 0.06 |
| 5 | $\delta_{MP}$ | 0.01 | 0.03 | 0.22 | 0.83 | -0.05 | 0.06 | 0.02 |
| 6 | $(\sigma_{LH} - \sigma_{LL}){:}\delta_{MP}$ | -0.05 | 0.04 | -1.25 | 0.21 | -0.14 | 0.03 | 0.00 |
| 7 | $(\sigma_{HL} - \sigma_{LH}){:}\delta_{MP}$ | 0.01 | 0.08 | 0.09 | 0.93 | -0.15 | 0.16 | 0.01 |
| 8 | $(\sigma_{HH} - \sigma_{HL}){:}\delta_{MP}$ | 0.06 | 0.10 | 0.58 | 0.56 | -0.14 | 0.25 | 0.02 |
| 9 | $\delta_{EP}$ | -0.06 | 0.12 | -0.52 | 0.60 | -0.29 | 0.17 | 0.01 |
| 10 | $(\sigma_{LH} - \sigma_{LL}){:}\delta_{EP}$ | 0.24 | 0.33 | 0.73 | 0.47 | -0.41 | 0.88 | 0.01 |
| 11 | $(\sigma_{HL} - \sigma_{LH}){:}\delta_{EP}$ | -0.07 | 0.34 | -0.21 | 0.83 | -0.74 | 0.60 | 0.01 |
| 12 | $(\sigma_{HH} - \sigma_{HL}){:}\delta_{EP}$ | -0.16 | 0.33 | -0.50 | 0.62 | -0.81 | 0.48 | 0.05 |
| 13 | log(Trial) | 1.72 | 0.05 | 33.82 | 0.00 | 1.62 | 1.82 | 0.56 |

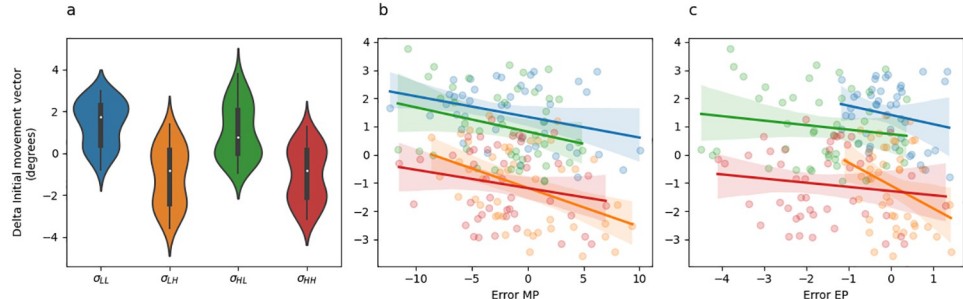

**Fig 13. Experiment 3 change in initial movement vector and linear regression fits.** (a) Violin plot depicting the distribution of mean changes in initial movement vector across all adaptation phase trials of the experiment separately for each midpoint/endpoint uncertainty combination, colour coded as per Fig 11a. The inset of each violin shows a box plot in which the white dot indicates the median data value, the black box spans the 25% to 75% percentiles, and the whiskers extend to the most extreme data points. (b) Scatter plot showing the mean change in initial movement vector as a function of error experienced at midpoint. Point and line colour indicates uncertainty level. (c) Scatter plot showing the mean change in initial movement vector as a function of error experienced at endpoint on the previous trial. The lines in panel B and C indicate fitted simple linear regression lines. These regression lines do not correspond to the coefficients included in Table 11) and are included only as a visual aid.

We also fit a regression model treating *change* in initial movement vector as the observed variable. Predictor variables for this model were identical to those described above but with no trial predictor. The best fitting betas along with their 95% confidence intervals is shown by the orange lines in Fig 12b. The model was statistically significant (Adjusted $R^2 = 0.507$, $F_{(11,166)} = 17.56$, $p < .001$).

A strong intuition for what this analysis ought to reveal can be built with careful inspection of Fig 13a, which depicts change in initial movement colour coded by the sensory uncertainty at midpoint on the previous trial. Here it is clear that change in initial movement vectors inversely tracks sensory uncertainty levels at endpoint. This result can be seen in the regression results by noting that change in initial movement vectors were significantly smaller following $\sigma_{HH}$ and $\sigma_{LH}$ trials than they were following $\sigma_{HL}$ and $\sigma_{LL}$ trials (Table 11, row 2–4).

Further intuitions can be built by examining Fig 13b, which depicts the change in initial movement vectors as a function of midpoint error. The lines in this plot are colour coded by

**Table 11. Experiment 3 regression results for predicting change in initial movement vector from error and sensory uncertainty terms.** These results correspond to the orange confidence intervals displayed in Fig 12. The *coef* column contains $\beta$ coefficients, the *se* column contains standard errors of these coefficients, the *T* column contains corresponding t-statistic, the *pval* column contains corresponding p-values, the *CI[2.5%]* and *CI[97.5%]* columns give the 95% confidence interval, and the *relimp* column gives the corresponding relative importance.

| row | names | coef | se | T | pval | CI[2.5%] | CI[97.5%] | relimp |
|---|---|---|---|---|---|---|---|---|
| 1 | $\beta_0$ | 0.47 | 0.21 | 2.25 | 0.03 | 0.06 | 0.88 | NaN |
| 2 | $\sigma_{LH} - \sigma_{LL}$ | -2.49 | 0.27 | -9.36 | 0.00 | -3.02 | -1.97 | 0.23 |
| 3 | $\sigma_{HL} - \sigma_{LH}$ | 3.52 | 0.62 | 5.69 | 0.00 | 2.30 | 4.74 | 0.05 |
| 4 | $\sigma_{HH} - \sigma_{HL}$ | -2.94 | 0.79 | -3.70 | 0.00 | -4.50 | -1.37 | 0.08 |
| 5 | $\delta_{MP}$ | -0.24 | 0.06 | -4.28 | 0.00 | -0.35 | -0.13 | 0.05 |
| 6 | $(\sigma_{LH} - \sigma_{LL}){:}\delta_{MP}$ | -0.03 | 0.09 | -0.37 | 0.71 | -0.21 | 0.14 | 0.00 |
| 7 | $(\sigma_{HL} - \sigma_{LH}){:}\delta_{MP}$ | -0.34 | 0.16 | -2.09 | 0.04 | -0.66 | -0.02 | 0.01 |
| 8 | $(\sigma_{HH} - \sigma_{HL}){:}\delta_{MP}$ | 0.27 | 0.20 | 1.30 | 0.19 | -0.14 | 0.67 | 0.02 |
| 9 | $\delta_{EP}$ | 0.58 | 0.24 | 2.38 | 0.02 | 0.10 | 1.05 | 0.02 |
| 10 | $(\sigma_{LH} - \sigma_{LL}){:}\delta_{EP}$ | -0.27 | 0.69 | -0.39 | 0.70 | -1.62 | 1.09 | 0.01 |
| 11 | $(\sigma_{HL} - \sigma_{LH}){:}\delta_{EP}$ | 1.33 | 0.71 | 1.88 | 0.06 | -0.07 | 2.72 | 0.01 |
| 12 | $(\sigma_{HH} - \sigma_{HL}){:}\delta_{EP}$ | -0.83 | 0.68 | -1.21 | 0.23 | -2.18 | 0.52 | 0.06 |

sensory uncertainty trial type. Fig 13c shows essentially the same information but for endpoint instead of midpoint error.

First note that every line across both of these plots has a negative slope indicating a general trend for both error sources to drive changes in initial movement vector. This is reflected in Table 11, rows 5 and 9 which show that both error at midpoint ($\delta_{MP}$) and error at endpoint ($\delta_{EP}$) are significant. Note that since the coefficient for midpoint error is negative, it drives changes in initial movement veector that reduce midpoint error on the next trial. However, note that the coefficient for endpoint error is positive, indicating that it drives changes in initial movement vector that increase error at endpoint on the next trial. Here we elect to not speculate too deeply into this finding given that the relative importance of these two terms was only 0.05 and 0.02.

Importantly, if sensory uncertainty at endpoint scales the response to error, then the slope of the blue and green lines ($\sigma_{LL}$ and $\sigma_{HL}$) should be the steepest, and the slope of the orange and red lines ($\sigma_{LH}$ and $\sigma_{HH}$) should be shallowest. Neither Fig 13b nor Fig 13c show this pattern. See Table 11 rows 6–8 for statistics corresponding to the interaction between sensory uncertainty with midpoint error, and rows 10–12 for statistics corresponding to the interaction between sensory uncertainty and endpoint error. Of these, only the $(\sigma_{HL} - \sigma_{LH}):\delta_{MP}$ term was significant and the relative importance of this term was only 0.01. On the other hand, the sum of the relative importance of sensory uncertainty terms was 0.36.

Overall, the results of this regression show that while both sensory uncertainty and movement error influence feedforward adaptation, they do so independently of each other. Furthermore, the influence of sensory uncertainty appears to outweigh the influence of movement error. Finally, like before, we find no evidence that sensory uncertainty scales the response to movement error.

**Feedback integration.** Fig 14a shows feedback integration (i.e., the difference between endpoint and midpoint hand angle) as a function of sensory uncertainty at midpoint, and Fig 14b shows feedback integration as a function of error at midpoint coloured by sensory

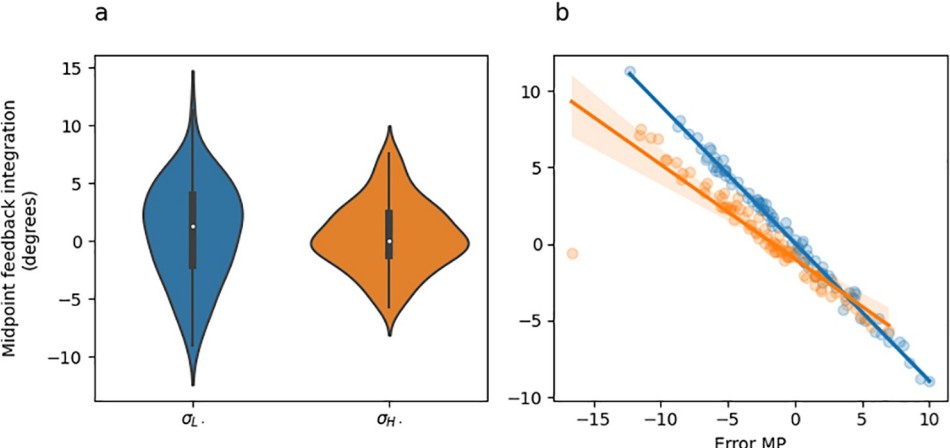

**Fig 14. Experiment 3 feedback integration (endpoint hand angle—initial movement vector).** (a) Violin plot depicting the distribution of mean feedback integration across all adaptation phase trials of the experiment separately for each midpoint uncertainty level. The inset of each violin shows a box plot in which the white dot indicates the median data value, the black box spans the 25% to 75% percentiles, and the whiskers extend to the most extreme data points. (b) Scatter plot showing the mean feedback integration as a function of error experienced at midpoint. Lines indicate fitted linear regression lines, corresponding to the coefficients included in Table 12, row 4). Point and line colour indicates uncertainty level.

**Table 12. Experiment 3 regression results for predicting feedback integration (endpoint hand angle—initial movement vector) from error and midpoint sensory uncertainty terms.** The *coef* column contains $\beta$ coefficients, the *se* column contains standard errors of these coefficients, the *T* column contains corresponding t-statistic, the *pval* column contains corresponding p-values, the *CI[2.5%]* and *CI[97.5%]* columns give the 95% confidence interval, and the *relimp* column gives the corresponding relative importance.

| row | names | coef | se | T | pval | CI[2.5%] | CI[97.5%] | relimp |
|---|---|---|---|---|---|---|---|---|
| 1 | $\beta_0$ | -0.46 | 0.07 | -6.44 | 0.0 | -0.61 | -0.32 | NaN |
| 2 | $\sigma_{MP}$ | -1.04 | 0.14 | -7.26 | 0.0 | -1.33 | -0.76 | 0.01 |
| 3 | $\delta_{MP}$ | -0.76 | 0.01 | -50.92 | 0.0 | -0.79 | -0.73 | 0.88 |
| 4 | $\sigma_{MP}{:}\delta_{MP}$ | 0.28 | 0.03 | 9.41 | 0.0 | 0.22 | 0.34 | 0.05 |

uncertainty. The main pattern observed in Fig 14 is that there are no significant differences in overall feedback integration depending on sensory uncertainty level (panel A), but there are large differences in how feedback integration responds to midpoint error depending on sensory uncertainty level (i.e., slope differences in panel B).

We formalised these observations by fitting a regression model treating the difference between endpoint and midpoint hand angle as the observed variable. Predictor variables were the error experienced at midpoint, the sensory uncertainty experienced at midpoint, and the interaction between these two terms. In contrast to the regression models reported for feedforward adaptation, all error and sensory uncertainty predictors were taken from the *current trial*. This regression was statistically significant (Adjusted $R^2 = 0.94$, $F_{(3,174)} = 925.1$, $p < .001$). Beta coefficient estimates and corresponding statistics are listed in Table 12.

The most important result from this analysis is that the response of feedback integration to increasing midpoint error (i.e., the slopes in Fig 14b) was significantly greater (i.e., steeper slope) for low sensory uncertainty at midpoint than it was for high uncertainty at midpoint (Table 12 row 4).

Overall, the pattern of feedback integration seen in Experiment 3 are qualitatively identical to those observed in Experiment 1 and 2. Both are consistent prior studies showing that that sensory uncertainty has an error-scaling effect on feedback integration [8, 10].

## Model-based results

Experiments that study feedforward adaptation in the absence of feedback integration have shown that sensory uncertainty scales an error-driven adaptation. Our experiments—which all provide a short 100 ms window of task-relevant midpoint feedback and therefore induce both feedback integration and feedforward adaptation—clearly show that while both movement error and sensory uncertainty influence feedforward adaptation, they appear to exert their influence independently of each other. Although it seems unlikely, it is possible that our data can be accounted for by a state-space model that embodies the classic view that sensory uncertainty scales the response to error. Another possibility is that error-driven motor adaptation drives the slow envelope of improving performance over the course of the experiment, and the effect of sensory uncertainty is to punctuate this envelope with abrupt changes (boosts or dips) that do not depend on the magnitude or direction of the experienced error.

Here we explore the ability of two classes of models to account for our data. The first class assumes that sensory uncertainty interacts with the updating of an adaptive learning process. This class of models contains the *error-scaling*, *retention-scaling*, and *bias-scaling* models described below. Of these, the *error-scaling* model embodies the classic view that sensory uncertainty scales an error-driven update. The second class of model assumes that sensory uncertainty interacts with an additional aiming process that is memory-less (i.e., it does not

retain any value from one trial to the next and is instead completely determined by the experienced sensory uncertainty on the previous trial). This class of models contains the *state-aim-scaling* and *output-aim-scaling* models described below. This model class is in line with the idea that sensory uncertainty transiently activates explicit aiming. However, there is noting intrinsic to this model that demands the aiming be driven by an explicit process (see the "Adaptation vs aiming" discussion section).

Each of these models makes importantly different assumptions about the role of sensory uncertainty on feedforward adaptation in the presence of feedback integration. The feedforward adaptation component of all five models was based on a standard linear dynamical systems model of sensorimotor adaptation [28]. This type of model assumes that internal state variables which map desired motor goals to motor plans are updated on a trial-by-trial basis according to three factors: (1) an error term that determines how internal states are updated after a movement error is detected, (2) a retention term that determines how quickly the internal state returns to baseline, and (3) a bias term that determines the baseline mapping that will be returned to in the absence of sensory input.

The error-scaling model assumes that the error term is inversely scaled by the level of sensory uncertainty. The retention-scaling model assumes that the retention term is inversely scaled by the level of sensory uncertainty. The the bias-scaling model assumes that the bias term is inversely scaled by the level of sensory uncertainty. The state aim-scaling model is equivalent to the bias-scaling model with zero retention and zero error sensitivity. In contrast to all other models, the output-aim-scaling model assumes that the output motor command—as opposed to some aspect of the internal state—is inversely scaled by sensory uncertainty. For all models, four parameters ($\gamma_L$, $\gamma_M$, $\gamma_H$ and $\gamma_\infty$) encode the magnitude of scaling applied on low uncertainty, medium uncertainty, high uncertainty, and no-feedback trials.

Sensory uncertainty influences feedback integration in the same way across all five models: error experienced at reach midpoint generates a feedback motor command in the opposite direction of the sensed movement error with a magnitude that inversely scales with the level of sensory uncertainty of that signal. The magnitude of this scaling is captured by four parameters ($\eta_L$, $\eta_M$, $\eta_H$ and $\eta_\infty$).

We fit single-state and two-state variants of each of the models just described. As suggested by this nomenclature, single-state model variants assume a single state variable determines the mapping from motor goal to motor plan. Two-state variants assume this mapping is determined by two state variables, one fast but labile and the other slow but stable [29]. In our two-state models, uncertainty scaling was applied only to the fast state variable.

Finally, we fit two different versions of the two-state models that differed in the bounds that the parameters were allowed to take. In one version, these parameter bounds lead the internal state variables to take negative values. In the other version, bounds were constrained such that the internal state variables were not allowed to take negative values. See supplemental figures for slow and fast-state model fits (S1–S30 Figs), as well as distributions of best-fitting parameters (S1–S3 Tables) as described in the State-space modelling section of the Methods and Materials. The BIC values across all models and experiments are presented in Fig 15.

We also performed a model rank analysis in which, for each participant within each experiment, we ordered the models from best fitting (smallest BIC) to worst fitting (largest BIC). We then counted the number of participants, best fit (rank 1), second best fit (rank 2), third best fit (rank 3) etc. for each model. The resulting counts are shown in Fig 16. This figure clearly reveals that in Experiment 1 the *State-aim-scaling-two-state* and *Output-aim-scaling-two-state* models provide the first and second best fits to the data. The non-negative versions of these models, in addition to the *bias-scale-two-state* model, also provide the third, fourth, and fifth best account of the data. A very similar pattern is observed in Experiment 2 with the *State-*

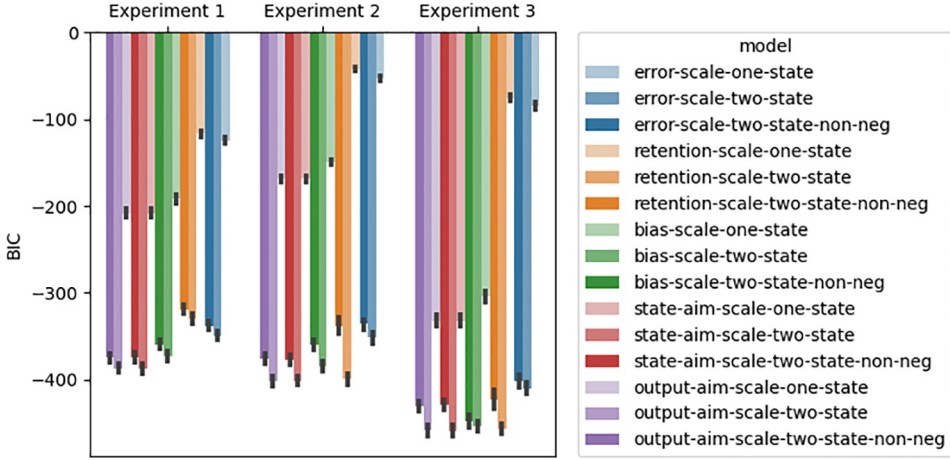

**Fig 15. Bar graph depicting model BIC values for Experiments 1,2 and 3.** Error-scaling model variants in blue. Retention-scaling variants in orange. Bias-scaling variants in green. State-aim variants in red. Output-aim variants in purple. Opacity indicate model sub-type: Non-negative two-state models have 100% opacity. Two-state models have 50% opacity and one-state models have 25% opacity. Error-bars represent 95% confidence intervals.

*aim-scaling-two-state* and *Output-aim-scaling-two-state* models providing the majority of best and second best fits. However, here here the *retention-scale-two-state* model provided the best fit for 4 participants, the second best fit for 3 particpants, and the third best for 12 participants. Here, the *bias-scale-two-state* model provided the fourth best fit for 16 participants. The results for Experiment 3 are slightly more varied with best and second best fits being split between the *State-aim-scaling-two-state*, *Output-aim-scaling-two-state*, *retention-scale-two-state*, and *bias-scale-two-state* models. These models in addition to the *bias-scale-two-state-non-neg* are also the third best fitting models. All variants of the *error-scaling* models did not perform well by this analysis, being ranked between 6 and 10 across all experiments. Additionally, Figs 17, 18 and 19 illustrate the predicted behaviour from each model class in comparison to the behaviour observed in humans for Experiment 1, Experiment 2, and Experiment 3, respectively.

We performed one paired sample t-tests per experiment to compare BIC values for two-state vs one-state model variants. These tests indicated that in all experiments the BIC values of the two-state models was significantly better (more negative) than that of the one-state models

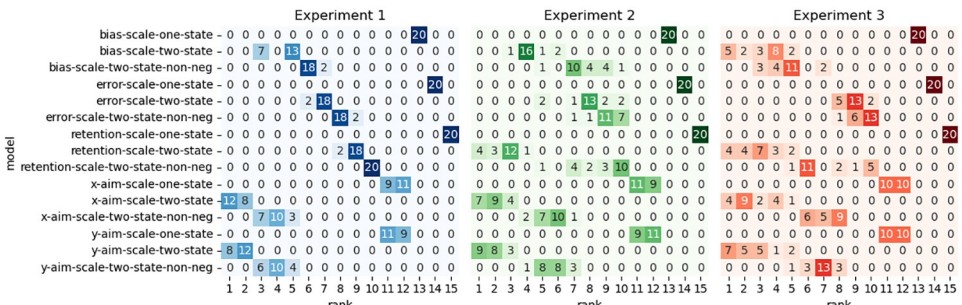

**Fig 16. Model rank analysis showing the number of particpants that were best fit (rank 1), second best fit (rank 2), third best fit (rank 3) etc.** by a model of each type. Results for Experiment 1 are shown on the left using shades of blue, results for Experiment 2 are shown in the middle using shades of green, and results for Experiment 3 are shown on the right using shades of red. Deeper colours indicate a greater count.

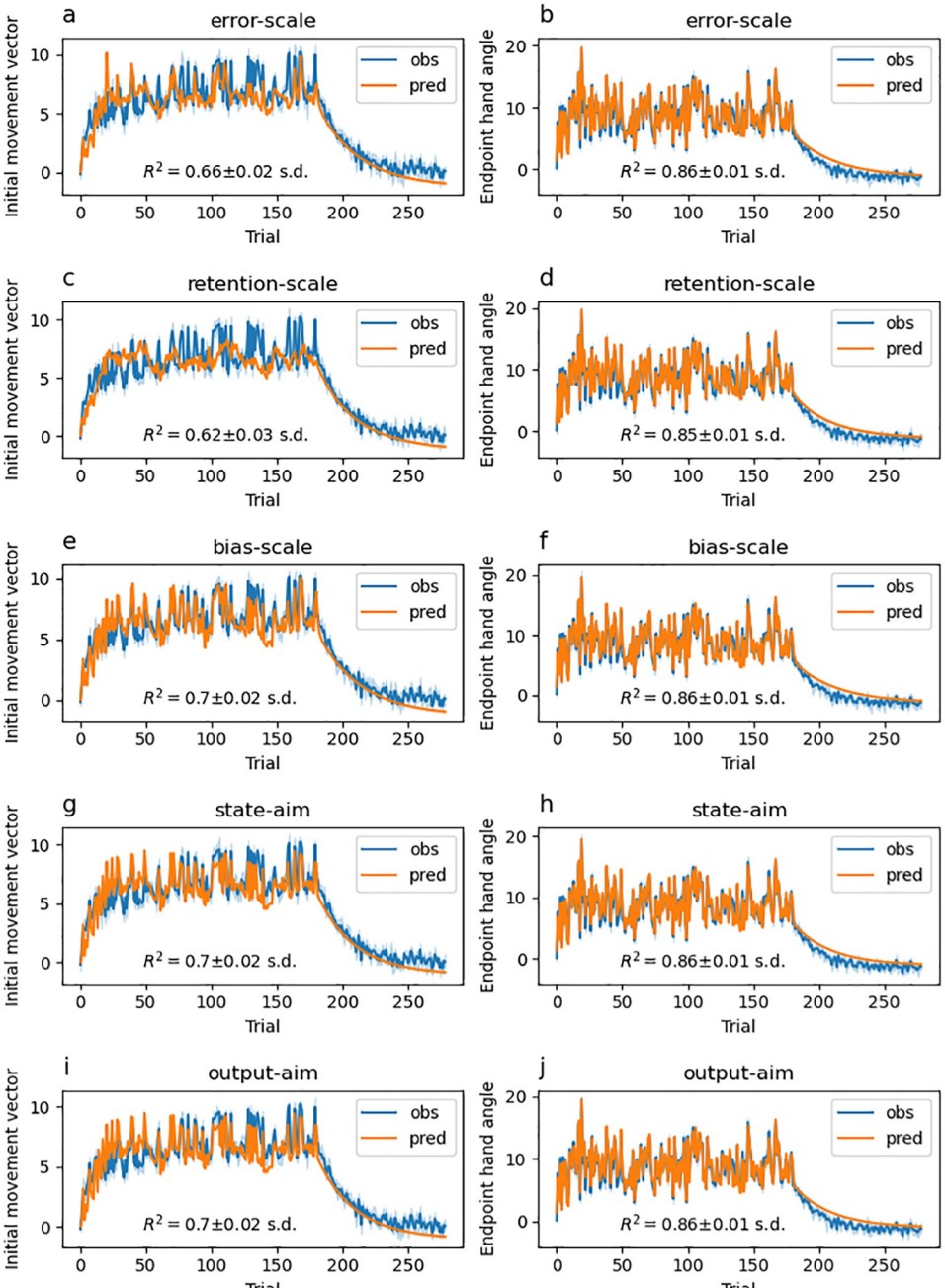

**Fig 17. Experiment 1 model fits.** Left column shows initial movement vectors averaged across participants overlaid with the average full model prediction of the (A) Two-state error-scaling model. (C) Two-state retention-scaling model. (E) Two-state bias-scaling model. (G) Two-state state-aim-scaling model (I) Two-state output-aim-scaling model. Right column shows corresponding model fits to endpoint hand-angles. Here, human performance averaged across participants is shown in blue. Model predictions in orange. Fit lines and $R^2$ values represent average of models fit to individual subjects.

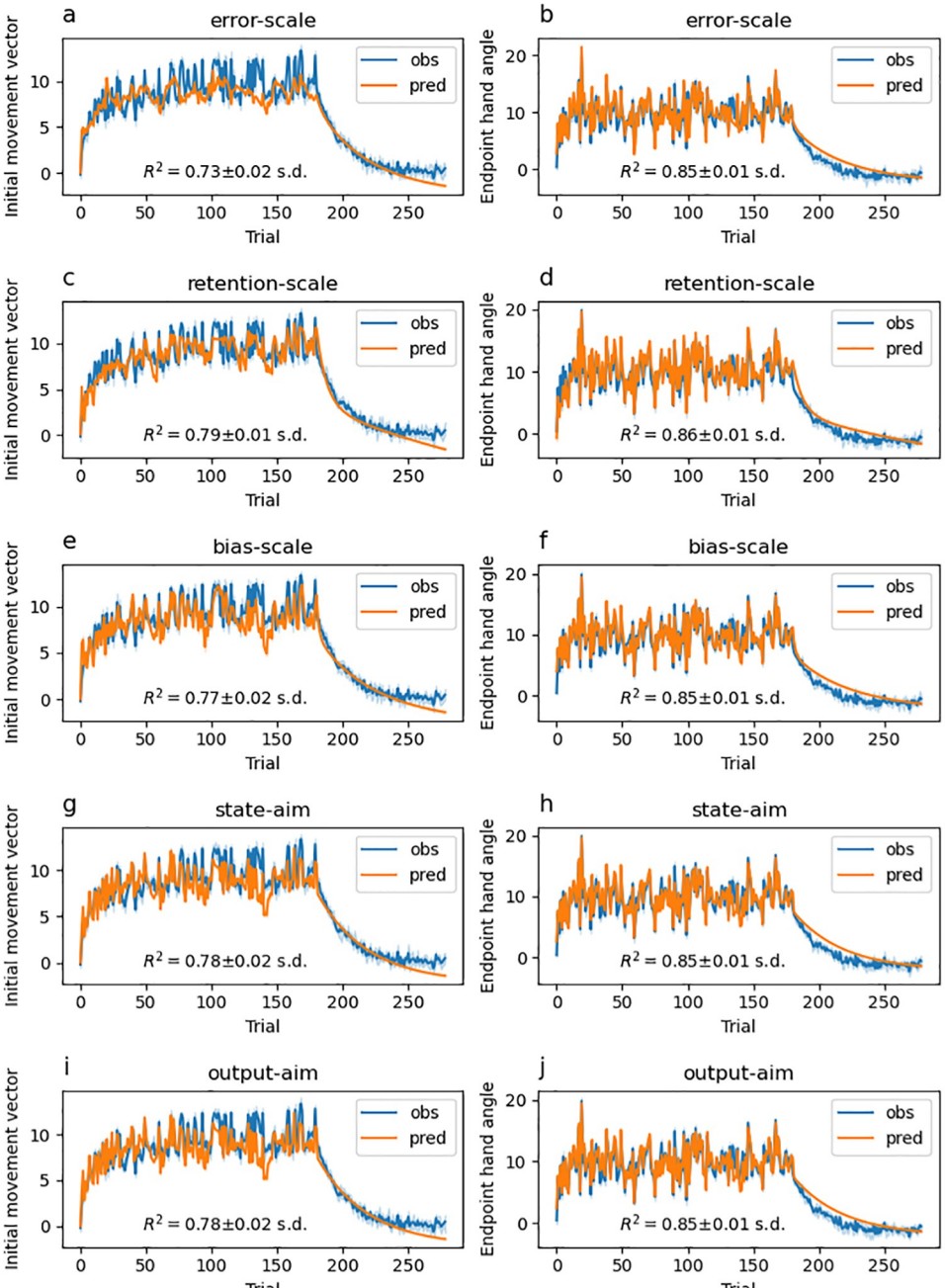

**Fig 18. Experiment 2 model fits.** Left column shows initial movement vectors averaged across participants overlaid with the average full model prediction of the (A) Two-state error-scaling model. (C) Two-state retention-scaling model. (E) Two-state bias-scaling model. (G) Two-state state-aim-scaling model (I) Two-state output-aim-scaling model. Right column shows corresponding model fits to endpoint hand-angles. Here, human performance averaged across participants is shown in blue. Model predictions in orange. Fit lines and $R^2$ values represent average of models fit to individual subjects.

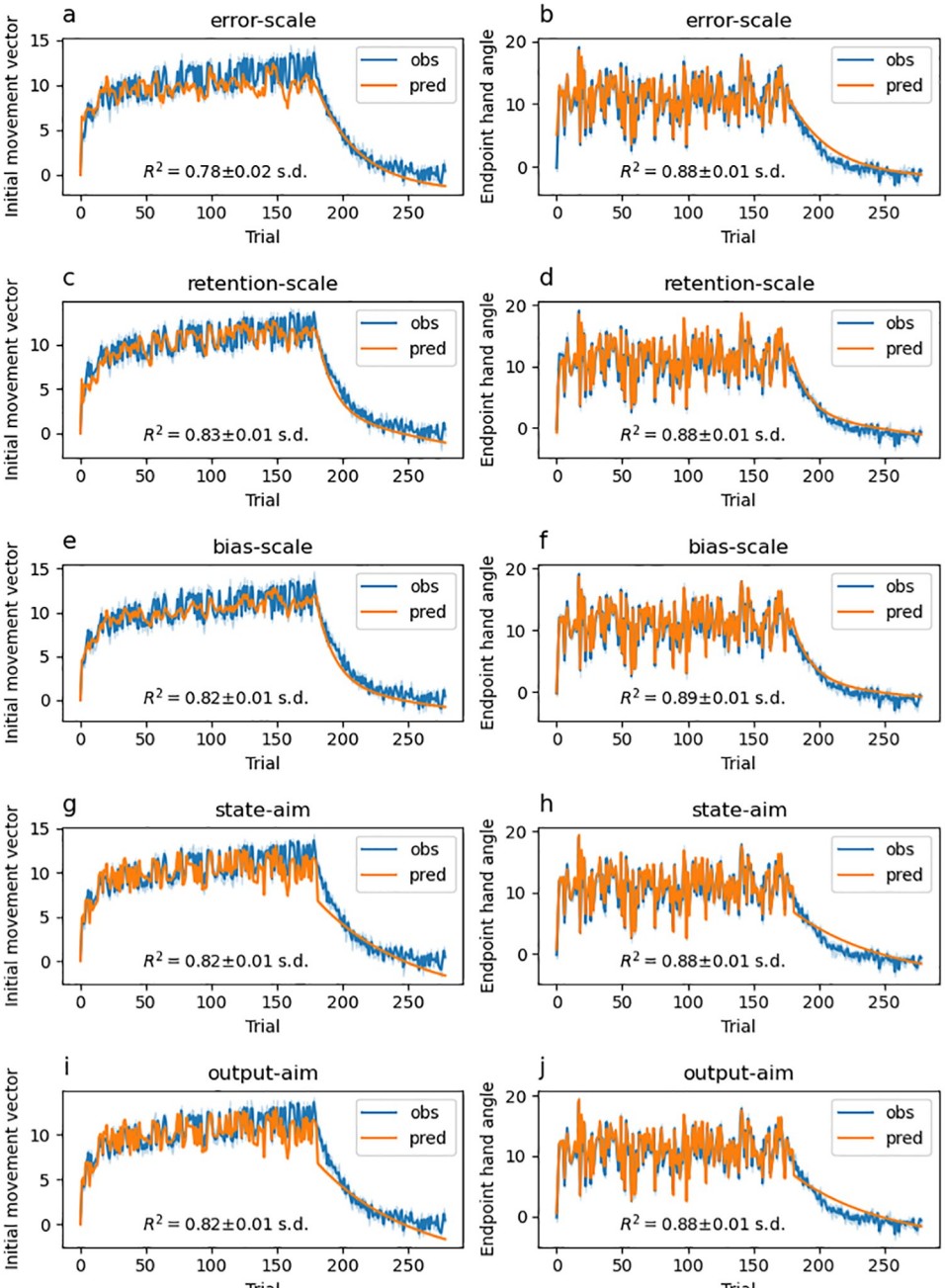

**Fig 19. Experiment 3 model fits.** Left column shows initial movement vectors averaged across participants overlaid with the average full model prediction of the (A) Two-state error-scaling model. (C) Two-state retention-scaling model. (E) Two-state bias-scaling model. (G) Two-state state-aim-scaling model (I) Two-state output-aim-scaling model. Right column shows the corresponding model fits to endpoint hand-angles. Here, human performance averaged across participants is shown in blue. Model predictions in orange. Fit lines and $R^2$ values represent average of models fit to individual subjects.

with a large effect size indicating a substantial difference between the two model types (Experiment 1: $t(19) = 68.68$, $p < 0.001$, $g = 16.77$; Experiment 2: $t(19) = 106.66$, $p < 0.001$, $g = 27.51$; Experiment 3: $t(19) = 90.04$, $p < 0.001$, $g = 17.06$). In Experiment 1, the mean BIC score for the two-state model was -359.41 (SD = 12.05) and for the one-state model was -169.77 (SD = 10.02). In Experiment 2, the mean BIC score for the two-state model was -372.48 (SD = 11.19) and for the one-state model was -116.42 (SD = 6.43). In Experiment 3, the mean BIC score for the two-state model was -437.17 (SD = 12.24) and for the one-state model was -225.94 (SD = 12.03).

We also performed paired sample t-tests per experiment to compare BIC values for the different model classes (i.e., error-scaling, retention-scaling, bias-scaling, state-aim-scaling, and output-aim-scaling). The state-aim-scaling (mean BIC = -380.96, SD = 12.64) and output-aim-scaling (mean BIC -380.95 SD = 12.63) models provided significantly better fits than any other model in Experiment 1, but could not be distinguished from each other (see S1 Table).

The same pattern was also found in Experiment 2, with state-aim-scaling (mean BIC = -389.00, SD = 12.81) and output-aim-scaling (mean BIC -388.88 SD = 12.64) models provided significantly better fits than any other model (see S2 Table). These models again could not be distinguished from each other.

In Experiment 3, the error-scaling model performed significantly worse than all other models, but the remaining models were all indistinguishable from each other (see S3 Table).

In summary, two-state models outperform one-state models, state-aim-scaling and output-aim-scaling models are the most commonly preferred model, and error-scaling models are never preferred. Although the aim-scaling models provides the best fits, all other two-state models nonetheless also provide good fits (i.e., high $R^2$ values). Consequently, any inferences drawn from the best-fitting model should be taken lightly, and future research is needed for more definitive model selection.

## Discussion

The current study is the first to examine how sensory uncertainty influences feedforward adaptation and feedback integration when they co-occur (but see our discussion of Körding and Wolpert [8] in the "Divergence from existing work" subsection below). In line with previous research, we find that the extent to which sensory feedback is integrated into an ongoing reach is inversely scaled by its level of uncertainty regardless of the presence or absence of feedforward adaptation [8–10, 30]). However, in sharp contrast to previous studies—all of which have found that sensory uncertainty inversely scales an error-driven update [8, 9, 11, 12, 14–17]—we show that the level of sensory uncertainty experienced in the previous trial punctuates a slow envelope of error reduction with large and abrupt changes to initial movement vectors that are insensitive to the magnitude and direction of the sensed movement error on the previous trial. Our results are highly novel and prompt important questions for future sensorimotor learning research to address.

### Divergence from existing work

Standard models of motor learning assume a linear relation between adaptation rate and error size [14, 28, 31–35], and the influence of sensory uncertainty on this process has been thought to inversely scale the error-driven update [11–13, 36, 37] (i.e., our error-scaling model). These results have often been interpreted through the lens of Bayesian [36] or other optimality frameworks such as that of Kalman filters [38]. These frameworks assume that as sensory uncertainty increases the motor system should limit adaptation in response to observed errors because they likely reflect higher sensory noise (to which adaptation would be sub-optimal) instead of

actual changes in the external environment (to which adaptation would be optimal). Consequently, when sensory uncertainty is high, the motor system should adapt less to a given error.

The observed pattern in our data is inconsistent with this characterization. In particular, while both sensory uncertainty and movement error influences feedforward adaptation, they do so independently of each other (but see the "Adaptation vs aiming" subsection below). For instance, the change in initial movement vectors after high sensory uncertainty trials consistently led to increased error on the subsequent trial, regardless of the magnitude and direction of the movement error experienced in the previous trial. This finding is supported by our regression analyses, which revealed that the interaction terms between sensory uncertainty and error were consistently non-significant or exhibited a pattern inconsistent with error-scaling.

Motivated by this apparent insensitivity to error, we developed a set of models in which sensory uncertainty has no effect on the error-driven component of feedforward adaptation. In the bias-scaling model, sensory uncertainty scales a constant bias term in the feedforward update. In the retention-scaling model, sensory uncertainty scales the rate that the system returns to baseline. In the state-aim scaling model, sensory uncertainty scales a bias term in the absence of retention or error updating, and in the output-aim scaling model, sensory feedback directly influences the feedforward motor output. In every experiment—and even in every participant—all of these "error-independent" models provided better fits than the error-scaling model. This is a significant point of divergence from the existing literature.

## Adaptation versus aiming

Our results can be viewed from at least two perspectives. From one perspective, the initial movement vectors can be seen as reflecting the current state of feedforward adaptation. In this view, adaptation following low sensory uncertainty trials behaves as expected, leading to adjustments that tend to reduce errors in subsequent trials. However, trials following greater sensory uncertainty induce adjustments that increase errors in subsequent trials. It is this latter pattern of behaviour that is seemingly problematic for existing models of sensorimotor learning.

An alternative perspective suggests that the initial movement vectors result from two factors. First, they reflect the current state of feedforward adaptation, which dictates how actions are executed. Second, they also carry the influence of explicit aiming strategies that dictate the selection of what action to execute [24, 39–41]. From this perspective, explicit aiming may drive the abrupt changes in initial movement vectors following low sensory uncertainty trials. For example, low sensory uncertainty trials might offer a clear signal for participants to notice the mismatch between their aiming point and where the cursor actually lands. This observation might lead to the generation of a hypothesis about the direction and magnitude of the perturbation. Following trials with greater sensory uncertainty, participants' confidence in this hypothesis may be eroded due to poor sensory feedback. As a result, participants might attempt to reach straight to the target (or as straight as their current level of adaptation allows) to obtain a better explicit estimate of the true perturbation.

This possibility is consistent with the pattern of initial movement vectors observed during the washout phase of all three experiments. Specifically, the initial movement vectors begin their decay back to baseline from the current state of the higher uncertainty trials, not from the level of the lower uncertainty trials. Furthermore, our modeling results strongly favored the state-aim-scaling and output-aim-scaling models, which are also consistent with this possibility. Recall that these models assume an aiming process with no retention from one trial to the next and solely rely on the sensory uncertainty experienced in the previous trial.

On the other hand, our proposed explanation for the involvement of explicit aiming in the observed data remains highly speculative. For instance, it is unclear why high uncertainty trials would prompt participants to abandon their previously successful explicit strategy, only to suddenly revert to the same strategy just a few trials later. Additionally, it is puzzling why almost all participants would invoke the same explicit strategy in the same way during these trials. This observation seems inconsistent with explicit aiming, as one would expect humans to devise different coping strategies.

Furthermore, it is important to acknowledge that our study was not designed to distinguish between implicit feedforward adaptation and explicit aiming processes. Additionally, all the models we tested yielded relatively high $R^2$ values, and this suggests that models which do not resemble explicit aiming can still effectively account for our data. Ultimately, further research is needed to understand how sensory uncertainty affects the interplay between feedforward adaptation and explicit aiming.

### Key paradigm differences

Our study closely resembles the design of Körding and Wolpert (2004) [8], which is one of the seminal studies in establishing error-scaling as a model of sensory uncertainty on feedforward adaptation. However, a crucial distinction lies in our specific focus on the first 180 trials of adaptation, while their study examined behavior after 2000 trials of adaptation had already taken place. This difference in design reflects the divergence in research questions between our two studies. Körding and Wolpert aimed to investigate if participants had learned a prior distribution of perturbations and whether they would incorporate new information into that prior in a Bayesian manner. Therefore, exposing participants to thousands of trials of the perturbation served as a practical way to impose a prior onto their subjects. They used the interplay between sensory uncertainty and feedback integration as a readout of what participants believed about the perturbation and how they updated these beliefs. In contrast, our study emphasizes understanding the interplay of feedback integration and feedforward adaptation as participants encounter the perturbation for the first time (e.g., when errors are likely to be relatively large and frequent [13, 42]). In principle, we could have directly compared our results to the early trials in Körding and Wolpert's study. However, regrettably, they did not report this data, leaving us unable to make a direct comparison between our feedforward adaptation results and their findings.

Since the seminal work of Körding and Wolpert (2004) [8], several other studies have investigated the influence of sensory uncertainty on feedforward adaptation, and all of these have supported an error-scaling model. Our study differs from these in a few ways. First, we interleaved different uncertainty conditions psuedo-randomly across trials whereas most other relevant studies used blocked designs [9, 13, 17, 43]. However, since Wei and Körding [12] also used a trial-interleaved design and found support for an error-scaling model, this is unlikely to be an important driver in our results.

Two remaining paradigm differences seem most compelling. In particular, our study is the first to investigate the influence of sensory uncertainty on feedforward adaptation when feedback integration and feedforward adaptation co-occur. In contrast, most existing studies only deliver feedback at movement endpoint (but see our discussion of Körding and Wolpert (2004) [8] at the top of this section). In doing so, they largely prevent corrections from occurring during the movement [11–13, 15–17]. This is the fundamental design feature we set out to manipulate in this study.

A final possibility is that our results may reflect the effect of sensory uncertainty on an explicit aiming process (as discussed above in the "Adaptation versus aiming" subsection),

whereas some existing studies have used designs that limit the influence of explicit aiming. For example, one study employed a zero-mean variable perturbation ([12]) and another employed task-irrelevant clamped feedback ([13]. Ultimately, adjudicating between an explanation based on the co-occurrence of feedback integration and feedforward adaptation processes and one based on changes to explicit aiming processes will require further research.

## Conclusion

Both the degree to which sensory feedback is integrated into an ongoing movement and the degree to which movement errors drive changes in feedforward motor plans have been shown to scale inversely with sensory uncertainty. Yet, little is known about how they respond to sensory uncertainty in real-world movement contexts where they co-occur. Here, we show that in this context, participants gradually adjust their movements from trial-to-trial in a manner that is well characterised by a slow and consistent envelope of error reduction, but also exhibit large and abrupt changes in their initial movement vectors that correlate with the degree of sensory uncertainty present on the previous trial yet are insensitive to the magnitude and direction of the sensed movement error. This may be seen as contextual alteration to the adaptation of feedforward motor plans (i.e., changes in how actions are executed) or as contextual alteration to the selection of what action to execute (e.g., aiming). In either case, our results prompt important questions for current models of sensorimotor learning under uncertainty and open up interesting new avenues for future exploration in the field.

## Materials and methods

### Participants

A total of 60 naive participants (32 males, 28 females, age 17–33 years) with normal or corrected to normal vision and no history of motor impairments participated in the experimental study. All participants gave written informed consent before the experiment and were either paid and recruited from the Macquarie University Cognitive Science Participant Register or were Macquarie University undergraduates participating for course credit. Neither written nor verbal consent was obtained from parents or guardians of participants aged 17 years (n = 2) because these participants were deemed capable of providing their own consent according to our ethics protocol. All experimental protocols were approved by the Macquarie University Human Research Ethics Committee (protocol number: 52020339922086). Participants were randomly assigned to one of three experiments (n = 20 per experiment). Sample sizes were consistent with field-standard conventions for visuomotor adaptation experiments [44–46].

### Experimental apparatus

A unimanual KINARM endpoint robot (BKIN Technologies, Kingston, Ontario, Canada) was utilized in the experiments for motion tracking and stimulus presentation (Fig 1). The KINARM has a single graspable manipulandum that permits unrestricted 2D arm movement in the horizontal plane. A projection-mirror system enables presentation of visual stimuli that appear in this same plane. Participants received visual feedback about their hand position via a cursor (solid white circle, 2.5 mm diameter) controlled in real-time by moving the manipulandum. Mirror placement and an opaque apron attached just below the subject's chin ensured that visual feedback from the real hand was not available for the duration of the experiment.

## General experimental procedure

Participants performed reaches with their dominant (right) hand from a starting position located at the center of the workspace (solid red circle, 0.5cm in diameter) to a single reach target (solid green circle, 0.5 cm in diameter) located straight ahead (0˚ in the frontal plane) at a distance of 10 cm. When participants moved the cursor within the boundary of the start target its colour changed from red to green and the reach target appeared, indicating the start of a trial. participants were free to reach at any time after the start target colour changed. Participants first completed a 20 trial baseline phase during which veridical online feedback was provided. Immediately following baseline, a 180 trial adaptation phase was completed. During the adaptation phase, once the cursor exited the start target, cursor feedback was extinguished and rotated counterclockwise (to the left) of the true hand position by an amount drawn at random on each trial from a Gaussian distribution with a fixed mean of 12˚ and standard deviation of 4˚. Random trial-by-trial perturbations, the order of which was trial-matched across all participants, were applied to prevent completely predictable movements during the adaptation phase and to probe the effect of sensory uncertainty at a trial-by-trial resolution. Participants were instructed to use cursor feedback to guide their reaches whenever it was available, and to move their hand straight through the target as accurately as possible. There were no breaks between phases, and transitions between phases were not explicitly signaled to participants in any way.

Depending upon the specific experiment (see descriptions below for details), displaced cursor feedback was provided at reach midpoint (100ms duration) and/or at endpoint (100ms duration), or withheld altogether. To help guide the participant's hand back to the starting position, a green ring centered over the starting position appeared with a radius equal to the distance between the hand and starting position. Once the participant's hand was 1 cm from the starting position, the ring was removed and cursor feedback was reinstated.

To investigate the effect of sensory uncertainty on feedback integration and feedforward adaptation, information provided about the visuomotor perturbation (true cursor position) was manipulated in the following way. One of four visual uncertainty levels ($\sigma_L$, $\sigma_M$, $\sigma_H$, $\sigma_\infty$) were selected and applied on a given trial according to the specific experimental protocol, with the trial sequence matched across participants. In the zero uncertainty condition ($\sigma_L$), feedback was a single white circle (0.5 cm in diameter; 5.73˚ arc-angle at midpoint, 2.86˚ at endpoint), identical to the initial cursor. In the moderate uncertainty condition ($\sigma_M$), feedback was one of 10 randomly generated point clouds comprised of 50 small translucent white circles (0.1 cm in diameter) distributed as a two-dimensional Gaussian with a standard deviation of 0.5cm (5.73˚ arc-angle at midpoint, 2.86˚ at endpoint), and a mean centered over the true (perturbed) cursor position on the current trial. In the high uncertainty condition ($\sigma_H$), everything was the same as the moderate uncertainty condition ($\sigma_M$) except that the point clouds had a SD of 1 cm (11.47˚ arc-angle at midpoint, 5.73˚ at endpoint). In the unlimited uncertainty condition ($\sigma_\infty$), no feedback was provided at all.

Immediately following the adaptation phase, participants experienced a 100 trial washout phase during which no cursor feedback was provided. The maximum allowable time to complete a reach was 1000 ms. Irrespective of the cursor's position, if participants did not cross the lower bound of the end target radius (9.5cm) the trial would time out and restart. If reaches exceeded the time limit or did not cross the lower bound of the target, the trial was repeated.

## Experiment 1

All four feedback uncertainty types ($\sigma_L$, $\sigma_M$, $\sigma_H$, $\sigma_\infty$) were applied on 25% of trials (45 trials each) at midpoint only (Fig 2a). No feedback was provided at endpoint. The order of

uncertainty conditions and perturbation values were randomised and trial-matched across all participants.

### Experiment 2

The protocol employed in Experiment 2 was identical to Experiment 1 except that both midpoint and endpoint feedback were provided on each trial (Fig 2b). Midpoint and endpoint feedback had matched uncertainty levels.

### Experiment 3

Experiment 3 consisted of four trial types (Fig 2c). Trial type 1 consisted of low uncertainty midpoint and low uncertainty endpoint feedback ($\sigma_{LL}$). Trial type 2 consisted of low uncertainty midpoint and high uncertainty endpoint feedback ($\sigma_{LH}$). Trial type 3 consisted of high uncertainty midpoint and low uncertainty endpoint feedback ($\sigma_{HL}$). Trial type 4 consisted of high uncertainty at midpoint and high uncertainty at endpoint feedback ($\sigma_{HH}$). Each of the four trial types occurred on 25% of trials (45 trials each).

### Data analysis

Movement kinematics including hand position and velocity were recorded for all trials using BKIN's Dexterit-E experimental control and data acquisition software (BKIN Technologies). Data was recorded at 200 Hz and logged in Dexterit-E. Custom scripts for data processing were written in MATLAB (R2013a). Data analysis and model fitting was done in Python (3.7.3) using the numpy (1.19.2) [47], SciPy (1.4.1) [48], pandas (1.1.3) [49], matplotlib (3.3.2) [50], and pingouin (0.3.11) [51] libraries. We report $\Delta BIC$ and compare the *BIC* distributions for model comparison via Dunnett's post-hoc test and correct for multiple comparisons using the Bonferroni correction.

A combined spatial- and velocity-based criterion was used to determine movement onset, movement offset, and corresponding reach endpoints [52, 53]. Movement onset was defined as the first point in time at which the movement exceeded 5% of peak velocity after leaving the starting position. Movement offset was similarly defined as the first point in time at which the movement dropped below 5% of peak velocity after a minimum reach of 9.5 cm from the starting position in any radial direction, and reach endpoint was defined as the (*x*, *y*) coordinate at movement offset. The optimal movement trajectory is a straight line between the start and end targets. Accordingly, the initial movement vector (IMV) is the angular difference between the optimal vector and the movement vector at movement onset and endpoint hand angle is the angular difference between the optimal vector and the movement vector at movement offset. During the adaptation phase, initial movement vectors were analyzed to explore the influence of sensory uncertainty on feedforward adaptation. Endpoint hand angles were analysed to explore feedback integration. During the no-feedback washout phase, initial movement vectors were analysed to investigate adaptation aftereffects.

### Statistical modelling

To quantify the effect of sensory uncertainty on feedforward adaption, we fit a regression model to the data from each of our experiments. We treated initial movement vector as the observed variable. Predictor variables were trial, the error experienced at midpoint, the error experienced at endpoint, the sensory uncertainty experienced at midpoint and/or endpoint *on the previous trial*, and the interaction between the error terms and the sensory uncertainty terms. We omitted the terms for error experienced at endpoint and the corresponding

interaction terms in our analysis of Experiment 1 because no endpoint feedback was provided on any trial in this experiment.

We used backward difference coding to enter sensory uncertainty (which we treat as ordinal) into this regression model. According to this coding scheme, the performance with sensory uncertainty at one level is compared with performance when sensory uncertainty is at the previous level. Thus, the regression models contain beta coefficients that capture the difference between (1) moderate uncertainty and low uncertainty, (2) high uncertainty and moderate uncertainty and (3) unlimited uncertainty and high uncertainty. This also applies to the interaction terms. In addition, we also fit a regression model using the *change* in initial movement vector from trial to trial as the observed variable. This regression used all of the same terms as the regression just described with the exception that we did not include trial as a predictor. If sensory uncertainty influences feedforward adaptation by scaling the feedforward controller's response to experienced errors, we should expect to find significant interaction terms between sensory uncertainty and error in our regression model.

We took a similar approach to determine how sensory uncertainty influences feedback integration. We fit regression models using endpoint hand angle as the observed variable. Predictor variables were the error experienced at midpoint, the sensory uncertainty experienced at midpoint *on the current trial*, and the interaction of these two terms. As before, we used backwards difference coding to enter sensory uncertainty into the regression model. Hence, the interaction terms in this model also indicate whether or not sensory uncertainty scales feedback integration. Finally, we also fit a regression model using the difference between endpoint hand angle and initial movement vector (i.e., feedback integration) as the observed variable. This regression used all of the same terms as the regression just described, with the exception that we did not include trial as a predictor.

In all models for which trial number was taken to be a predictor, trial number was transformed using the natural logarithm. This has the effect of turning our non-linear adaptation curves into straight lines, and thereby makes linear regression a more appropriate analysis tool for our research question. The models were fit to the group averaged data using ordinary least squares to obtain best-fitting parameter estimates. We also report the relative importance of each regressor following the methods developed in [54].

## State-space modelling

To characterize how participants' reaching behaviour changed over time, we also fit three different linear dynamical system models to our data. At a coarse-grained level, each model is characterised by the following features:

- A feedforward motor plan is computed at movement onset that is an attempt to reach in a straight line from the starting position to the target location, and a feedback motor command is computed at movement midpoint that is an attempt to correct the ongoing movement for any error experienced at midpoint.

- Feedforward motor plans are adapted on a trial-by-trial basis using both the error experienced at midpoint and the error experienced at endpoint as learning signals.

- The gain applied to feedback corrections is similarly adjusted on a trial-by-trial basis, but is sensitive only to the error experienced at endpoint.

- The sensory uncertainty experienced at midpoint and/or endpoint modulates the between-trial feedforward update and the within-trial feedback correction, but not the between-trial feedback gain update (for the sake of simplicity).

The feedforward adaptation component of all three models is based on simple discrete-time linear dynamical systems—so-called state-space models [28]. The simplest version of these models assumes that an internal state variable $x$ maps desired motor goals to motor plans $y$, and that $x$ is updated on a trial-by-trial basis in response to sensory feedback about movement error. The update to $x$ has (1) an error term that determines how the internal state is updated after a movement error is detected, (2) a bias term that determines the baseline mapping that will be returned to in the absence of sensory input and (3) a retention term that determines how quickly the internal state returns to baseline after sensory feedback about error is removed. This arrangement is encapsulated in the following equations:

$$\delta(n) = y^*(n) - y(n) \tag{1}$$

$$x(n + 1) = \beta x(n) + \alpha \delta(n) + \lambda \tag{2}$$

$$y(n) = x(n) + r(n) \tag{3}$$

where $n$ is the current trial, $\delta(n)$ is the error (i.e., the angular distance between the reach endpoint and the target location), $y^*(n)$ is the desired output (e.g., the angular position of the reach target), $y(n)$ is the motor output and corresponds to the angle of the movement that will be generated when trying to reach to the target (i.e., it is a readout of the sensorimotor state), $x(n)$ is the state of the system (i.e., the sensorimotor transformation), $\beta$ is a retention rate that describes how much is retained from the value of the state at the previous trial, $\alpha$ is a learning rate that describes how quickly states are updated in response to errors, $\lambda$ is a constant bias, and $r(n)$ is the imposed rotation.

Note that the bias term ($\lambda$) is applied in the state-update equation and not in the motor-output equation. In this form, the bias term will ultimately produce a stable bias unless it is acted upon by sensory uncertainty (as it does in the bias-scaling models—see below) in which case it will cause trial-to-trial changes in the underlying adapted states.

These models are sometimes equipped with a second internal state variable [29, 39] as follows:

$$\delta(n) = y^*(n) - y(n) \tag{4}$$

$$x_f(n + 1) = \beta_f x_f(n) + \alpha_f \delta(n) + \lambda \tag{5}$$

$$x_s(n + 1) = \beta_s x_s(n) + \alpha_s \delta(n) + \lambda \tag{6}$$

$$y(n) = x_f(n) + x_s(n) + r(n) \tag{7}$$

where $x_f$ is a *fast* state variable, $x_s$ is a *slow* state variable, $\beta_f < \beta_s$, and $\alpha_f > \alpha_s$. That is, feedforward adaptation is often assumed to arise from the combination of a slow-but-stable system and a fast-but-labile system. Previous studies have not clearly established the appropriateness of one-state versus two-state models for capturing how sensory uncertainty influences feedforward adaptation. Consequently, we explore both one-state and two-state model variants in this paper.

The simple state-space framework just described assumes motor output reflects the execution only of feedforward motor commands, whereas the behaviour observed in our experiments is also likely influenced by feedback motor commands. We therefore augment the simple state-space model as follows.

The total motor output of the model is defined at three discrete time points within each trial. We denote the time of reach initiation as $t_0$, the time of midpoint crossing as $t_{MP}$, and the time of endpoint crossing as $t_{EP}$. The total motor output on trial $n$ at any time $t$ denoted $y(n, t)$ is a combination of feedforward $y_{ff}(n)$ and feedback $y_{fb}(n, t)$ motor commands as follows:

$$y(n, t) = y_{ff}(n) + y_{fb}(n, t) \tag{8}$$

Note that feedforward motor output is not a function of time within a trial because we assume that the feedforward motor output is computed at $t_0$ and remains fixed throughout the rest of each trial. This is equivalent to assuming that the execution of the movement occurs too rapidly for new feedforward motor planning to influence the ongoing movement.

In the *single-state* models, the feedforward motor command $y_{ff}(n)$ is determined by a single internal state variable denoted by $x_{ff}(n)$ that maps the current movement goal to motor commands as follows:

$$y_{ff}(n) = x_{ff}(n) \tag{9}$$

In the *two-state* models, the feedforward motor command $y_{ff}(n)$ is determined by two internal state variables denoted by $x_{ff_f}(n)$ and $x_{ff_s}(n)$ that map the current movement goal to motor commands as follows:

$$y_{ff}(n) = x_{ff_f}(n) + x_{ff_s}(n) \tag{10}$$

At reach initiation, sensory feedback has not yet been provided so the feedback motor command is zero:

$$y_{fb}(n, t_0) = 0 \tag{11}$$

If sensory feedback is provided at midpoint, then the following sensory prediction error is experienced:

$$\delta(n, t_{MP}) = y(n, t_0) + r(n) \tag{12}$$

Here, $\delta(n, t_{MP})$ is the sensory prediction error, and $r(n)$ is the visuomotor rotation applied on trial $n$. Notice that the motor command issued at time $t_0$ is responsible for generating the sensory prediction error at time $t_{MP}$. In response to this sensory prediction error, the following compensatory feedback motor command is triggered:

$$y_{fb}(n, t_{MP}) = -x_{fb}(n)\delta(n, t_{MP})\boldsymbol{\eta}\boldsymbol{I}(n) \tag{13}$$

Here, $x_{fb}(n)$ is an internal state variable that represents the gain of the feedback controller, $\boldsymbol{\eta} = [\eta_0, \eta_M, \eta_H, \eta_\infty]$ is a row vector of free parameters encoding the sensory uncertainty of the midpoint feedback (one value for each possible level of sensory uncertainty), and $\boldsymbol{I}(n)$ is a column vector that indicates what level of midpoint uncertainty was present on trial $n$. Notice that the feedback motor command is just some fraction of the experienced error in magnitude and in the opposite direction—because of the leading negative sign—hence it serves to reduce movement error.

If endpoint sensory feedback is provided, the following sensory prediction error is experienced:

$$\delta(n, t_{EP}) = y(n, t_{MP}) + r(n) \tag{14}$$

Notice that the motor command issued at time $t_{MP}$ is responsible for generating the sensory prediction error at time $t_{EP}$. In the transition from trial $n$ to trial $n + 1$, the gain of the feedback

controller is updated in response to this sensory prediction error as follows:

$$x_{\text{fb}}(n+1) = \beta_{\text{fb}} x_{\text{fb}}(n) + \alpha_{\text{fb}} \delta(n, t_{\text{EP}}) \tag{15}$$

Note that we assume that updates to feedback gain are not sensitive to sensory uncertainty. Evidence that feedback controllers are well described by this process comes from studies of so-called gain adaptation [18–21, 55].

We built several models which fall into two classes of assumptions regarding how sensory uncertainty influences motor learning. The first class of models assumes that sensory uncertainty influences the updating of the adaptive learning process (i.e., it takes hold somewhere in the state-update equation). This class of models includes the *error-scaling*, *retention-scaling*, and *bias-scaling* models. *Error-scaling* models assume that sensory uncertainty scales the contribution of the error term (e.g., $\alpha\delta(n)$ in Eq 2), *retention-scaling* models assume that sensory uncertainty scales the contribution of the retention term (e.g., $\beta x(n)$ in Eq 2), and *bias-scaling* models assume that sensory uncertainty scales the contribution of the bias term (e.g., $\lambda$ in Eq 2).

The second class of models assumes that sensory uncertainty influences an aiming process that contains no retention from one trial to the next (i.e., it is memoryless) and is completely determined by the sensory uncertainty experienced on the previous trial. This class of models contains the *state-aim-scaling* and *output-aim-scaling* models. *state-aim-scaling* models are equivalent to bias-scaling models but with error and retention terms set to zero, and *output-aim-scaling* models assume that sensory uncertainty scales the motor output ($y(n)$ in Eq 3). This model class is consistent with the idea that sensory uncertainty triggers explicit aiming (see the "Adaptation vs aiming" discussion section) though the aiming process need not strictly be explicit.

In the *two-state* version of these models, we assume that sensory uncertainty only influences $x_{\text{ff}_f}$. That is, we assume that $x_{\text{ff}_s}$ is independent of sensory uncertainty. In particular, the feed-forward internal state $x_{\text{ff}_s}$ is updated between trials in response to the sensory prediction errors experienced at midpoint and at endpoint as follows:

$$\begin{aligned}
x_{\text{ff}_s}(n+1) = \quad & \beta_{\text{ff}_s} x_{\text{ff}_s}(n) + \\
& \alpha_{\text{ff}_s} \delta(n, t_{\text{MP}}) + \\
& \alpha_{\text{ff}_s}[\delta(n, t_{\text{EP}}) - y_{\text{fb}}(n, t_{\text{MP}})]
\end{aligned} \tag{16}$$

Here, $\alpha_{\text{ff}_s}$ is a learning rate parameter and $\beta_{\text{ff}_s}$ is a retention parameter, both bound between [0, 1]. Notice that the feedback command issued at midpoint is taken to be an error signal in these equations—in the term $[\delta(n, t_{\text{EP}}) - y_{\text{fb}}(n, t_{\text{MP}})]$—which is a common assumption in models that join feedforward and feedback control [18–21].

**Error-scaling models.** The uncertainty of sensory feedback influences the update to $x_{\text{ff}}$ in the *single-state error-scaling* model by acting as a gain on the learning rate $\alpha_{\text{ff}}$. The update is given by:

$$\begin{aligned}
x_{\text{ff}}(n+1) = \quad & \beta_{\text{ff}} x_{\text{ff}}(n) + \\
& [\gamma][\boldsymbol{\nu} \boldsymbol{I}_{\text{MP}}(n)][\alpha_{\text{ff}} \delta(n, t_{\text{MP}})] + \\
& [1 - \gamma][\boldsymbol{\nu} \boldsymbol{I}_{\text{EP}}(n)]\big[\alpha_{\text{ff}}[\delta(n, t_{\text{EP}}) - y_{\text{fb}}(n, t_{\text{MP}})]\big] + \\
& \lambda
\end{aligned} \tag{17}$$

Here, $\boldsymbol{\nu(n)} = [\nu_0, \nu_M, \nu_H, \nu_\infty]$ is a row vector of free parameters (one value for each level of sensory uncertainty), with boundary conditions between [0, 1], to represent the scaling effect

of sensory uncertainty. $I_{\mathrm{MP}}(n)$ is a column vector that indicates the uncertainty of sensory feedback that was present on trial $n$ at midpoint, $I_{\mathrm{EP}}(n)$ is a column vector that indicates the uncertainty at endpoint, and $\gamma$ is a temporal discounting parameter, bound between [0, 1], that determines the relative weighting of midpoint versus endpoint feedback on the overall state update. For instance, if $\gamma > 0.5$, midpoint feedback drives the majority of the state update. If $\gamma < 0.5$, endpoint feedback drives the majority of the state update. Note that any interpretation of the temporal discount parameter is relevant only in the case of Experiment 3, which is the only paradigm that applies unmatched midpoint and endpoint feedback, a feature required to demarcate the effect of the learning rate parameter from the effect of the temporal discounting parameter. The constant bias term $\lambda$ is bound between [-10, 10]. The update to $x_{\mathrm{ff}_f}$ in the *two-state error-scaling* follows exactly the same equation, while the update to $x_{\mathrm{ff}_s}$ is denoted by Eq 16. The classic finding that sensory uncertainty inversely scales the magnitude of the error-driven component of the feedforward update [9, 11–13, 15–17] would be recapitulated here if (1) the error-scaling model provides the best-fitting account of our data, and (2) the best-fitting parameters were such that $v_0 > v_M > v_H > v_\infty$.

**Retention-scaling models.** The uncertainty of sensory feedback influences the update to $x_{\mathrm{ff}}$ in the *single-state retention-scaling* model by acting as a gain on the retention term $\beta_{\mathrm{ff}}$. The update is given by the following equations:

$$
\begin{aligned}
x_{\mathrm{ff}}(n+1) = \quad & [\gamma \boldsymbol{v} I_{\mathrm{MP}}(n) + (1-\gamma)\boldsymbol{v} I_{\mathrm{EP}}(n)][\beta_{\mathrm{ff}} x_{\mathrm{ff}}(n)] + \\
& [\alpha_{\mathrm{ff}} \delta(n, t_{\mathrm{MP}})] + \\
& [\alpha_{\mathrm{ff}}[\delta(n, t_{\mathrm{EP}}) - y_{\mathrm{fb}}(n, t_{\mathrm{MP}})]] + \\
& \lambda
\end{aligned}
\tag{18}
$$

All parameters, nomenclature and bounds are identical to those described above for the error-scaling model. The update to $x_{\mathrm{ff}_f}$ in the *two-state retention-scaling* model follows exactly the same equation, while the update to $x_{\mathrm{ff}_s}$ is denoted by Eq 16.

**Bias-scaling models.** The uncertainty of sensory feedback influences the update to $x_{\mathrm{ff}}$ in the *single-state bias-scaling* model by acting as a gain on the bias term $\lambda$. The update is given by:

$$
\begin{aligned}
x_{\mathrm{ff}_f}(n+1) = \quad & [\beta_{\mathrm{ff}_f} x_{\mathrm{ff}_f}(n)] + \\
& [\alpha_{\mathrm{ff}_f} \delta(n, t_{\mathrm{MP}})] + \\
& [\alpha_{\mathrm{ff}_f}[\delta(n, t_{\mathrm{EP}}) - y_{\mathrm{fb}}(n, t_{\mathrm{MP}})]] + \\
& [\gamma \boldsymbol{v} I_{\mathrm{MP}}(n) + (1-\gamma)\boldsymbol{v} I_{\mathrm{EP}}(n)]\lambda
\end{aligned}
\tag{19}
$$

All parameters and nomenclature are identical to those described above for the error-scaling model. The update to $x_{\mathrm{ff}_f}$ in the *two-state bias-scaling* follows exactly the same equation, while the update to $x_{\mathrm{ff}_s}$ is denoted by Eq 16.

**State-aim-scaling models.** The uncertainty of sensory feedback influences the update to $x_{\mathrm{ff}}$ in exactly the same fashion as it does for the *bias-scaling models*. The key difference is that the *state-aim-scaling* models have zero retention and zero error updating. The state update is given by:

$$
x_{\mathrm{ff}_f}(n+1) = [\gamma \boldsymbol{v} I_{\mathrm{MP}}(n) + (1-\gamma)\boldsymbol{v} I_{\mathrm{EP}}(n)]\lambda
\tag{20}
$$

All parameters and nomenclature are identical to those described above for the error-scaling model. The update to $x_{\text{ff}_f}$ in the *two-state state-aim-scaling* model follows exactly the same equation, while the update to $x_{\text{ff}_s}$ is denoted by Eq 16.

**Output-aim-scaling models.** In contrast to all other models described thus far, the uncertainty of sensory feedback does not influence the update to $x_{\text{ff}}$ at all in *output-aim-scaling models*. Rather, these models assume that the uncertainty of sensory feedback directly influences the feedforward motor output as follows:

$$y(n) = y_{ff}(n) + y_{fb}(n) + y_{aim}(n) \tag{21}$$

$$y_{aim}(n) = [\gamma \boldsymbol{v} \boldsymbol{I}_{\text{MP}}(n - 1) + (1 - \gamma)\boldsymbol{v}\boldsymbol{I}_{\text{EP}}(n - 1)]\lambda \tag{22}$$

All parameters and nomenclature are identical to those described above for the error-scaling model.

**Parameter estimation.** For each model, we obtained best-fitting parameter estimates on a per subject basis by minimising the following sum of squared error difference between the

**Table 13. State-space model parameter bounds.** Lower and upper values are indicated by (lb, ub), respectively. The nomenclature (-,-) indicates that this parameter was not present in the corresponding model. Blank entries indicate that the bounds were inherited from the one-state model.

| parameter | Error- and retention- scaling | | | Bias-scaling | | | State-aim- and output-aim- scaling | | |
|---|---|---|---|---|---|---|---|---|---|
| | one-state | two-state | non-neg | one-state | two-state | non-neg | one-state | two-state | non-neg |
| | (lb, ub) | (lb, ub) | (lb, ub) | (lb, ub) | (lb, ub) | (lb, ub) | (lb, ub) | (lb, ub) | (lb, ub) |
| $\alpha_s$ | (-, -) | (0, 1) | | (-, -) | (0, 1) | | (-, -) | (0, 1) | |
| $\beta_s$ | (-, -) | (0, 1) | | (-, -) | (0, 1) | | (-, -) | (0, 1) | |
| $\lambda_s$ | (-, -) | (0, 0) | | (-, -) | (0, 0) | | (-, -) | (0, 1) | |
| $\alpha_f$ | (0, 1) | | | (0, 1) | | | (-, -) | | |
| $\beta_f$ | (0, 1) | | | (0, 1) | | | (-, -) | | |
| $\lambda_f$ | (-10, 10) | | (0, 10) | (-10, 10) | | (0, 10) | (-, -) | | (0, 10) |
| $\alpha_{fb}$ | (0, 1) | | | (0, 1) | | | (0, 1) | | |
| $\beta_{fb}$ | (-10, 10) | | | (-10, 10) | | | (-10, 10) | | |
| $x_{\text{fb init}}$ | (-2, 2) | | | (-2, 2) | | | (-2, 2) | | |
| $v_{ff1}$ | (0, 1) | | | (0, 1) | | | (0, 1) | | |
| $v_{ff2}$ | (0, 1) | | | (0, 1) | | | (0, 1) | | |
| $v_{ff3}$ | (0, 1) | | | (0, 1) | | | (0, 1) | | |
| $v_{ff4}$ | (0, 1) | | | (0, 1) | | | (0, 1) | | |
| $v_{fb1}$ | (0, 1) | | | (-1, 1) | | | (-20, 20) | | (0, 20) |
| $v_{fb2}$ | (0, 1) | | | (-1, 1) | | | (-20, 20) | | (0, 20) |
| $v_{fb3}$ | (0, 1) | | | (-1, 1) | | | (-20, 20) | | (0, 20) |
| $v_{fb4}$ | (0, 1) | | | (-1, 1) | | (0, 1) | (-20, 20) | | (0, 20) |
| $\gamma$ | (0, 1) | | | (0, 1) | | | (0, 1) | | |

observed and predicted midpoint and endpoint hand angles:

$$E = \sum_{i}^{N} [y_{\text{pred}}(i, t_{\text{MP}}) - y_{\text{obs}}(i, t_{\text{MP}})]^2 + \tag{23}$$

$$\sum_{i}^{N} [y_{\text{pred}}(i, t_{\text{EP}}) - y_{\text{obs}}(i, t_{\text{EP}})]^2 \tag{24}$$

Here, $N$ is the number of trials, $y_{\text{pred}}(i, t_{\text{MP}})$ and $y_{\text{pred}}(i, t_{\text{EP}})$ are the model predicted hand angle at midpoint and endpoint, respectively, on trial $i$, and $y_{\text{obs}}(i, t_{\text{MP}})$ and $y_{\text{obs}}(i, t_{\text{EP}})$ are the corresponding hand angles observed in a human participants. To find the parameter values that minimised $E$, we used the differential evolution optimization [56] method implemented in *SciPy* [48].

**Parameter bounds.** Table 13 shows the bounds under which parameter optimization was constrained. Note that we fit two different sets of bounds for the two-state model. The first, simply called *two-state*, allowed the internal state variables ($x_s$ and $x_f$) to sometimes take negative values. If the fast internal state variable ($x_f$) corresponds to an explicit aiming strategy, then owing to the inherent flexibility of such aiming strategies, these negative values do not seem particularly problematic or counter-intuitive. However, in our study at least, $x_f$ cannot be unambiguously linked to cognitive aiming strategies and may instead reflect the operation of an implicit adaptation system. In this case, negative internal state values may potentially be more biologically implausible. For example, what conditions would induce an implicit adaptation system to drive one state variable in a highly positive direction and the other slightly negative? Such a system could surely exist in principle, but given the state of knowledge in our field, it seems less parsimonious then a system for which both slow and fast internal state variables remain positive. For this reason we also fit a version of the two-state model with parameter bounds that ensured that both state variables remained positive. We call these *non-neg-two-state* models.

**Model comparison.** For each model, we computed the Bayesian Information Criterion (*BIC*) as follows:

$$BIC = n \, ln \, (1 - R^2) + k \, ln \, (n) \tag{25}$$

Here $k$ represents the number of model parameters for each model, $n$ represents the number of data points, and $R^2$ is the proportion of variance explained by the optimised model. Models with lower *BIC* value are preferred [57].

## Supporting information

**S1 Table. Experiment 1 two-state model comparison statistics.** Abbreviations are *std* is standard deviation, $T$ is the t-statistic, *dof* is degrees of freedom, *p-corr* is the p-value corrected for multiple comparisons, and *hedges* is Hedges *g*.
(PDF)

**S2 Table. Experiment 2 two-state model comparison statistics.** Abbreviations are *std* is standard deviation, $T$ is the t-statistic, *dof* is degrees of freedom, *p-corr* is the p-value corrected for multiple comparisons, and *hedges* is Hedges *g*.
(PDF)

**S3 Table. Experiment 3 two-state model comparison statistics.** Abbreviations are *std* is standard deviation, $T$ is the t-statistic, *dof* is degrees of freedom, *p-corr* is the p-value corrected for

multiple comparisons, and *hedges* is Hedges *g*.
(PDF)

**S1 Fig. Experiment 1: Model fit and optimized parameter distributions for the two-state error-scaling model.**
(TIF)

**S2 Fig. Experiment 1: Model fit and optimized parameter distributions for the two-state error-scaling non-negative λ model.**
(TIF)

**S3 Fig. Experiment 1: Model fit and optimized parameter distributions for the two-state retention-scaling model.**
(TIF)

**S4 Fig. Experiment 1: Model fit and optimized parameter distributions for the two-state retention-scaling non-negative λ model.**
(TIF)

**S5 Fig. Experiment 1: Model fit and optimized parameter distributions for the two-state bias-scaling model.**
(TIF)

**S6 Fig. Experiment 1: Model fit and optimized parameter distributions for the two-state bias-scaling non-negative λ model.**
(TIF)

**S7 Fig. Experiment 1: Model fit and optimized parameter distributions for the two-state output-scaling model.**
(TIF)

**S8 Fig. Experiment 1: Model fit and optimized parameter distributions for the two-state output-scaling non-negative λ model.**
(TIF)

**S9 Fig. Experiment 1: Model fit and optimized parameter distributions for the two-state aim-scaling model.**
(TIF)

**S10 Fig. Experiment 1: Model fit and optimized parameter distributions for the two-state aim-scaling non-negative λ model.**
(TIF)

**S11 Fig. Experiment 2: Model fit and optimized parameter distributions for the two-state error-scaling model.**
(TIF)

**S12 Fig. Experiment 2: Model fit and optimized parameter distributions for the two-state error-scaling non-negative λ model.**
(TIF)

**S13 Fig. Experiment 2: Model fit and optimized parameter distributions for the two-state retention-scaling model.**
(TIF)

**S14 Fig. Experiment 2: Model fit and optimized parameter distributions for the two-state retention-scaling non-negative $\lambda$ model.**
(TIF)

**S15 Fig. Experiment 2: Model fit and optimized parameter distributions for the two-state bias-scaling model.**
(TIF)

**S16 Fig. Experiment 2: Model fit and optimized parameter distributions for the two-state bias-scaling non-negative $\lambda$ model.**
(TIF)

**S17 Fig. Experiment 2: Model fit and optimized parameter distributions for the two-state output-scaling model.**
(TIF)

**S18 Fig. Experiment 2: Model fit and optimized parameter distributions for the two-state output-scaling non-negative $\lambda$ model.**
(TIF)

**S19 Fig. Experiment 2: Model fit and optimized parameter distributions for the two-state aim-scaling model.**
(TIF)

**S20 Fig. Experiment 2: Model fit and optimized parameter distributions for the two-state aim-scaling non-negative $\lambda$ model.**
(TIF)

**S21 Fig. Experiment 3: Model fit and optimized parameter distributions for the two-state error-scaling model.**
(TIF)

**S22 Fig. Experiment 3: Model fit and optimized parameter distributions for the two-state error-scaling non-negative $\lambda$ model.**
(TIF)

**S23 Fig. Experiment 3: Model fit and optimized parameter distributions for the two-state retention-scaling model.**
(TIF)

**S24 Fig. Experiment 3: Model fit and optimized parameter distributions for the two-state retention-scaling non-negative $\lambda$ model.**
(TIF)

**S25 Fig. Experiment 3: Model fit and optimized parameter distributions for the two-state bias-scaling model.**
(TIF)

**S26 Fig. Experiment 3: Model fit and optimized parameter distributions for the two-state bias-scaling non-negative $\lambda$ model.**
(TIF)

**S27 Fig. Experiment 3: Model fit and optimized parameter distributions for the two-state output-scaling model.**
(TIF)

**S28 Fig. Experiment 3: Model fit and optimized parameter distributions for the two-state output-scaling non-negative λ model.**
(TIF)

**S29 Fig. Experiment 3: Model fit and optimized parameter distributions for the two-state aim-scaling model.**
(TIF)

**S30 Fig. Experiment 3: Model fit and optimized parameter distributions for the two-state aim-scaling non-negative λ model.**
(TIF)

## Author Contributions

**Conceptualization:** Christopher L. Hewitson, David M. Kaplan, Matthew J. Crossley.

**Data curation:** Christopher L. Hewitson, David M. Kaplan, Matthew J. Crossley.

**Formal analysis:** Christopher L. Hewitson, David M. Kaplan, Matthew J. Crossley.

**Funding acquisition:** Christopher L. Hewitson, David M. Kaplan, Matthew J. Crossley.

**Investigation:** Christopher L. Hewitson, David M. Kaplan, Matthew J. Crossley.

**Methodology:** Christopher L. Hewitson, David M. Kaplan, Matthew J. Crossley.

**Project administration:** Christopher L. Hewitson, David M. Kaplan, Matthew J. Crossley.

**Resources:** Christopher L. Hewitson, David M. Kaplan, Matthew J. Crossley.

**Software:** Christopher L. Hewitson, David M. Kaplan, Matthew J. Crossley.

**Supervision:** David M. Kaplan, Matthew J. Crossley.

**Validation:** Christopher L. Hewitson, David M. Kaplan, Matthew J. Crossley.

**Visualization:** Christopher L. Hewitson, David M. Kaplan, Matthew J. Crossley.

**Writing – original draft:** Christopher L. Hewitson, David M. Kaplan, Matthew J. Crossley.

**Writing – review & editing:** Christopher L. Hewitson, David M. Kaplan, Matthew J. Crossley.

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
