## [Decision Letter · Decision Letter 0]

18 Oct 2022

Dear Dr Crossley,

Thank you very much for submitting your manuscript "Sensory uncertainty punctuates motor learning independently of movement error when both feedforward and feedback control processes are engaged" for consideration at PLOS Computational Biology.

As with all papers reviewed by the journal, your manuscript was reviewed by members of the editorial board and by several independent reviewers. The reviewers raised a number of significant issues in the manuscript. In particular, they had major concerns about the validity of the computational model and the ultimate interpretation of the findings. The reviewers were in agreement that the experiments and results were interesting and novel but that, at present, the paper falls short of providing a fully rigorous and convincing contribution to our understanding of motor adaptation. We would, however, be willing to consider a revised version of the manuscript in which the concerns raised by the reviewers are thoroughly addressed. Given the scope and nature of the concerns raised by the reviewers, the revisions would likely need to be extensive. Your revised manuscript will be sent out to the reviewers again to be re-evaluated and we cannot make any decision on this submission until after that time.

Sincerely,

Adrian M Haith

Academic Editor

PLOS Computational Biology

Daniele Marinazzo

Section Editor

PLOS Computational Biology

Reviewer's Responses to Questions

**Comments to the Authors:**

Reviewer #1: review uploaded as attachment

Reviewer #2: Summary:

The study investigates how visual uncertainty of movement feedback affects motor adaptation when the feedback is provided during and after the movement. Traditional motor adaptation studies on sensory uncertainty focus on how adaptation evolves between trials in a feedforward fashion. Instead, the paper studies a combined case when in-movement sensory integration and between-trial adaptation co-occur. With three behavioral experiments, the authors found that randomly presented high-uncertainty trials (implemented by modifying visual feedback during and after movements) lead to abrupt changes in feedforward adaptation, which are dependent on uncertainty level but not on movement error size. Model comparisons revealed that scaling the retention or trial-by-trial bias instead of the error adaptation rate can better explain the data. These findings are drastically different from those in the literature, where sensory uncertainty has been repetitively shown to affect the adaptation rate. 

 

The study is original, with a strong focus on probing its study aim. It is also novel in combining online movement feedback integration and offline motor adaptation. The technical part of modeling is also sound. However, the seemingly surprising results and their explanations invite scrutiny as the task design probably invokes unwanted effects, making their conclusions thus far ungrounded. 

 

Major concerns:

 

First, some important behavioral patterns and modeling results have not been fully analyzed or explained, preventing our thorough understanding of the interesting phenomena: 

1. The main selling point is that the mid-movement feedback enables a feedback integration component in addition to the feedforward adaptation. However, the paper only analyzed the initial movement direction to study feedforward adaptation. Is there any proof that feedback integration happens during the movement? Do we see a curved movement trajectory or a change in movement endpoint? How does this potential behavioral correction relate to the model-fitting results? It is well possible that the so-called feedback integration is negligible. And the influence of mid-movement feedback should be considered as part of feedforward adaptation, which only affects the next trial. This is a vital question before we try to understand what these peculiar findings mean.

2. A cross-experiment comparison is lacking. It is apparent that exp2 and 3 have larger adaptations than exp1. Do the three experiments have the same decay in the washout phase? How do the experiment differences inform the modeling? I see the slow state during the decay from all models is negative in all three experiments, up to -10 in exp3. This is rather surprising given the current understanding of the slow state, which probably reflects the implicit adaptation or the true re-calibration of the internal model. But it would never be negative. Note the decay portion of the data largely determines the model fitting.

3. The quick transitions between low and high uncertainty trials should be plotted better using other means, such as a scatter plot of uncertainty vs. movement direction (at the adaptation plateau). What do the adaptation changes look like for each kind of transition? Currently, this part of vital results is not even visible.  

4. What does the transition from the last adaptation trial to the first washout trial look like? This is another vital piece of missing information to unpack what constitutes the adaptation here. If the authors give clear instructions to drop the re-aiming strategy at the beginning of the washout, we should expect a sudden decrease in the movement direction. And, then the second hypothesis mentioned in the Discussion (strategic learning leads to fast changes during adaptation) can be examined.

5. Why does the state-scaling model win over the bias-scaling model in exp2 and 3 while we see the opposite in exp1? Note that the overall learning patterns are similar across experiments. Is it possible that the retention term and the bias term co-vary to generate the effect?

 

Second, though some of the behavioral data were not shown clearly in the paper, I think the task design has introduced some unwanted effects, which are not possibly modeled by the error-based model variants here.  

The perturbations are random rotation angles, ~ N(12,16) degrees. The gaussian point clouds are specified in cm (the paper should convert one or another to make a clear description of what the perturbation would look like). By my calculation, the mid-movement perturbation is, on average, centered at 1.06cm away from the desired straight line, and the endpoint perturbation is, on average, 2.12cm. The high-uncertainty dot cloud (with a SD of 1cm) would “touch” the straight line, thus giving people the impression that there is no perturbation (zero error). The medium-uncertainty dot cloud (with a SD of 0.5cm) would also touch the straight line if presented at the mid-movement. This is probably the reason that these trials lead to fast trial-by-trial changes, riding on top of a slow time-scale learning curve. These interleaved “target-hit” trials would affect strategic re-aiming and implicit adaptation. The former is driven by performance error (for the target hit, it is zero). The latter is also expected since, say, Richard Ivry group (Kim et al., 2019) have shown that touching the target, even barely without a direct hit, would effectively reduce the implicit adaptation. The mechanism for the damped implicit learning by this zero or reduced performance error (not sensory prediction error modeled here) is not clear, though reinforcement learning or motivational factors have been suggested. 

 

In other words, the study started off by implementing cloud dots to manipulate sensory uncertainty (precision) but triggered unwanted learning mechanisms that depend on performance error (bias). This target hit would be more severe for mid-movement feedback, which is the study subject of the paper. Thus, it is not fair, as claimed by the paper, that the sensory uncertainty at the midpoint, as opposed to the perturbation size, determines the feedforward adaptation here. Or, as it claims in the title, “sensory uncertainty punctuates motor learning independently of movement error…” It is the manipulation of the sensory uncertainty that accidentally nullifies the perceived error and thus leads to a temporary withholding of the adaptation process (note all these cloud-dot trials are interleaved with the small-uncertainty trials, which I believe is clearly perceived off the straight line). 

 

In this light, the so-called anti-adaptive effect, seen in angle changes between two cloud-dot trials (sigma_L to sigma_M, or sigma_M to sigma_H), is totally expected. Note the reduction of learning amounts to > 30% after single mid-movement feedback. This kind of fast change is indeed not expected by any model incorporating the effect of sensory uncertainty but is expected with a combination of a fast strategic re-aiming and damped implicit learning. Given these extremely volatile changes, the single-state models will not work, and the error-scaling model will not work either, given the normal range of learning rate. 

 

 

Minor:

 

I don’t understand why we can simply assume that the uncertainty only affects the fast state in the two-state variants of the models. I know that fast, abrupt adaptation changes are prominent in this data set (possibly due to the reasons I give above). But in the domain of motor adaptation, the slow process has been shown to be dependent on sensory uncertainty (e.g., an implicit adaptation that is supposedly governed by slow processes is affected by proprioceptive uncertainty). Theoretically, it is explicit learning (one of the fast processes) that is less affected by sensory uncertainty (precision) since the mean performance error (bias) matters more for strategical learning. Anyhow, in this sense, the results strongly suggest that the fast changes, which are uncertainty-dependent, reflect the explicit learning component. 

 

Figures 3, 5, and 7 can use a different color coding to distinguish the perturbation size better while making the zero perturbation a completely different color off the color-coding scale. 

 

Line614: x_fb(n) denotes the so-called feedback gain, but why feedback gain follows a state-space model as specified by eq.15? 

Eq14: y(n,t_MP) is unspecified. Note how this variable relates to eq.13 is also not given. 

 

As it currently stands, the model is poorly described. It is better to give a graphic illustration of how the modeled variables are related in this movement paradigm. Simply providing a list of equations would not give the readers a gist of what is modeled, especially when the model consists of two time scales, two time points of updating, annd two learning mechanisms (feedback and feedforward). To simplify the modeling part, I suggest only presenting the two-state model variants since they are well accepted as the default model for motor adaptation and also fit the data well. If possible, put the single-state model results in the supplementary and focus on comparing different scalings in the main text.  

 

Line632-635: eq.2, not 3?

 

eq.17: Is the gamma parameter necessary? Why do we need to assume that the mid-point feedback and the endpoint feedback compete?

 

Instead of giving out ANOVA results (Line185) to show significance, the author should provide condition means to give us a full picture of how this effect changes across uncertainty conditions.

 

Figures fonts are too small to read. 

 

Line349：unclear to me what the authors are trying to convey here. “Two uncertainty conditions” instead of “two high uncertainty conditions”?

 

Line393: Experiment 3

Reviewer #3: In this paper, Hewitson and colleagues investigate the effect of visual uncertainty on feedback and feedforward processes during motor adaptation. These influences had been studied separately for feedforward and feedback processes and they do it at the same time. The authors found that sensory uncertainty had the expected effect on feedback process (less correction when more uncertainty) but it had an unexpected effect on feedforward process (increasing uncertainty decreased adaptation).

I found the results intriguing and interesting. It is a pity that the authors did not investigate further whether the observed pattern could be due to the explicit component of adaptation. I am very skeptical of the modelling approach and there is little rationale for the different modelling choices. These and other comments are developed below.

Major comments:

1. The authors discuss the idea that the changes due to sensory uncertainty might be due to the explicit component of adaptation. They left it opened without real conclusion because they did not actually test it. This is in my opinion detrimental to the paper because it left it without a real conclusion. The results now are not really interpretable because we don’t know what the nature of the rapid changes is. I believe that this paper should test whether the explicit component of adaptation is responsible for the rapid shift or not. If it is due to explicit strategies, then it is not comparable to what happened to the feedback process which is likely implicit. If it is due to the explicit component of adaptation, then the model does not really make sense as explicit adaptation is driven by task performance error and not by sensory prediction error like the implicit component is. Yet, in current models in the paper, both states are driven by sensory prediction error.

2. I have several problems with the model. The model seems to be ad-hoc and not really supported by previous research or experimental observations:

a. The authors seem to confound task performance error and sensory prediction error. Sensory prediction error drives the implicit adaptation while the task performance error (midpoint or endpoint) drives the explicit component of adaptation and feedback responses. The model mixes the two continuously.

b. The authors model the feedback gains as following a state-space model (Eq.15). Where is the evidence for such a model? Feedback gains can be changed within a trial (work of Fred Crevecoeur on small and large targets) and are tuned continuously in function of the sensory feedback. I don’t think that the model would be able to capture that.

c. The model contains the idea that endpoint feedback is used to tune the gains on the next trial. Where is the evidence for that statement (Eq.16)? Eq.16 does not make sense to me. If it is a state-space, you need to update it once for midpoint and once for endpoint. Updating it based on two errors is weird.

d. Eq.16 also suggests that the sensory prediction error takes the feedback response into account. What is the evidence that this is actually the case?

e. Eq.17 suggests that sensory feedback from midpoint and endpoint is weighted to compute a weighted average error. What is the evidence that this weighting exists? In addition, I don’t see how this would generalize to a condition where visual feedback is continuously available.

f. I wonder where the parameter Lambda comes from. This bias-parameter is absent from previous versions of the two-state models and there is not real rationale to add it to the model. In addition, the authors decided to bound it [-10,10] without any further explanation.

g. In Eq.18 and 19, the authors use a weighted some of sensory uncertainties (on retention rate and lambda). This seems weird to me and I wonder what the rationale for that would be. Do the authors think that the influence of sensory uncertainty on the retention rate from midpoint and endpoint feedback is simply the average of their sensory uncertainty? That does not make sense.

In other words, I don’t think that the models are useful because they seem to have been built to specifically fit the results of the current experiments and will not be generalizable to any other experimental results.

3. There is a lack of information about the quality of the fit for individual participants. If Fig.4,6 and 8, R2 are given for all participants together but the quality of the fit for individual participants are never given. Coltman and Gribble (https://pubmed.ncbi.nlm.nih.gov/30840553/) showed that fits of two-state models are not very reliable for single participants while they were at the group level. Could the authors provide more information about the quality of the fit for the individual participants and a sensitivity analysis of the obtained parameters?

4. I found the title weird. There is no evidence in the paper that the sensory uncertainty does not punctuate motor learning. The slow system is not interrupted by the sensory uncertainty. None of the parameters become zero or so. The fast system is simply very volatile but this is added on top of a slow component that is still learning.

Minor comments:

1. Instructions delivered during the washout phase needs to be specified. It is written that the participants were asked to reach straight to the target but, despite this instruction, the pattern of after-effect looks very different than what others have reported with similar instructions (e.g. work of Taylor). It is also unclear how the participant made the difference between no-vision trials during the adaptation and during the washout. In other words, how was the waschout period signaled to the participant.

2. There are several problems with the report of the statistics. There are many p “>” 0.001, which should be “<”. Some statistics are weird. For instance, line 190-192, the statistics is wrongly reported. The p-value is p<0.001 and not p=0.99. This effect size is actually huge. Please, check all the statistics of the paper. Maybe use StatCheck?

3. The simulation with single state-space models is useless. From the experimental results, it is pretty clear that a model with a single state will never be able to reproduce such pattern.

**Have the authors made all data and (if applicable) computational code underlying the findings in their manuscript fully available?**

Reviewer #1: None

Reviewer #2: Yes

Reviewer #3: **No: **code is available, data is not

PLOS authors have the option to publish the peer review history of their article (what does this mean?). If published, this will include your full peer review and any attached files.

Reviewer #1: No

Reviewer #2: No

Reviewer #3: No
---

## [Decision Letter · Decision Letter 1]

23 May 2023

Dear Dr Crossley,

Thank you very much for submitting your manuscript "Error-independent effect of sensory uncertainty on motor learning when both feedforward and feedback control processes are engaged" for consideration at PLOS Computational Biology. As with all papers reviewed by the journal, your manuscript was reviewed by members of the editorial board and by two of the reviewers who reviewed the original submission. 

The reviewers appreciated the substantial efforts in revising the paper based on their prior comments, but also raised a number of remaining concerns, some conceptual in nature and others more technical. 

While I believe the reviewers have highlighted some excellent and very important points and have made numerous constructive suggestions, I actually don't find there to be many concerns in the reviews that I would consider essential to address before the paper can be published. The extensive comments from the reviewers are indicative that this is a very surprising and stimulating set of results. While there is undoubtedly scope to further improve the conceptual framing, the analytical approach, and other aspects of the paper, I also think it's clear that there is a lot of scope for further investigation here, and much of this feedback can be incorporated into future work, rather than being necessary to resolve ahead of publishing this paper.

Regarding the concerns related to interpretational issues. Some of these surround the role of implicit versus explicit processes in the results. I think it will be difficult and ultimately not really necessary to completely resolve these concerns here and now. While I agree that knowing for sure whether these effects relate to the "explicit" component would provide valuable context for the results, I don't think it would substantially impact or diminish the contributions of the present paper. There were also some concerns about terminology. However, I think this is inevitably a challenge when combining different strands of research. Provided the terminology is clearly defined (which I think it is), I think the readers will be able to follow the reasoning.

Regarding the technical issues relating to the modeling and analysis, the reviewers made numerous suggestions for the how the paper could be further improved in this regard. In particular, I think the model recovery analysis suggested by Reviewer 1 would be helpful in clarifying how clearly the various models can be distinguished - see also Wilson and Collins, eLife 2019 for a thorough discussion of this. Overall, however, I consider most of these points to be constructive suggestions for further improvement, rather than critical concerns.

In short, I feel the paper is very close to being acceptable, but the reviewers have made numerous constructive suggestions for how the paper might be further improved. I am therefore recommending a "Minor Revision" in which I encourage you to seriously engage with the reviewers' suggestions and make revisions where you feel these are appropriate and improve the paper. While I would like to see a point-by-point response to the reviewers' comments, I do not expect you to implement all the suggestions of the reviewers.

Sincerely,

Adrian M Haith

Academic Editor

PLOS Computational Biology

Daniele Marinazzo

Section Editor

PLOS Computational Biology

Reviewer's Responses to Questions

**Comments to the Authors:**

Reviewer #1: see attached comments

Reviewer #2: The revision answers some of my previous concerns and, more importantly, provides more experimental and modeling findings. Interestingly, all the reviewers pointed to one most probable explanation of the seemingly surprising data: the abrupt changes in initial movement direction are probably caused by rapid explicit strategy changes. The revision provides supporting evidence that the explicit learning, indirectly shown as the abrupt drop at the beginning of the washout, was predominant for the low-uncertainty condition. This condition showed the surprising abrupt "adaptation" on top of the slow adaptation with other uncertainty conditions. Furthermore, the modeling results also support the role of explicit strategy since the models with abrupt biasing terms (bias scale, state aim, and output aim models) outperform other models. It seems convincing that the main findings can be accounted for by the explicit error correction triggered by the low-uncertainty feedback. 

However, the authors still interpret the findings as supporting that the presence of feedback integration (aka., the midpoint feedback as opposed to the endpoint feedback that was used in previous studies) alters the effect of uncertainty on feedforward adaptation. I have concerns over the study rationale, the interpretation of the findings, and the way how the study relates to existing literature.

 About the relation between the current study and existing studies. The paper claims it is the first to examine sensory uncertainty when feedforward adaptation and feedback integration co-occur. This is an overstatement. Kording & Wolpert 2004 paper, the seminar paper the current study is based on, arguably combined these two processes. They made people adapt to a lateral shift (adaptation) and, on top of that, used midpoint feedback to probe the "feedback integration" to test within-trial response, just like the current paper did. Furthermore, it is not fair to say that the midpoint feedback is for feedback processes and the endpoint feedback is for feedforward processes. I appreciate that the authors give their definitions of feedback integration and feedforward adaptation in the text, but I respectfully disagree with these terminologies as it creates confusion. Endpoint feedback IS involved in feedback processes and feedback integration; it is just that its effect can only be seen in the next trial for reaching paradigms. Redefining the term will unnecessarily obscure the message of the paper. The authors can call the behavioral measures within-trial correction and cross-trial correction.

 The paper starts with the recognition that sensory uncertainty is believed to slow down adaptation. In the abstract, they mentioned, "Both the degree to which sensory feedback is integrated into an ongoing movement and the degree to which movement errors drive adaptive changes in feedforward motor plans have been shown to scale inversely with sensory uncertainty." However, references 15-22 (quoted on L65 in the Introduction) are not just for showing that sensory uncertainty slows down adaptation. Some quoted studies manipulated the variability of perturbation/error size, targeting uncertainty about perturbation size (or mapping uncertainty in Burge2008's terminology, or environmental/perturbation consistency in Herzfeld2014's terminology) but not about sensory uncertainty. The Kording2004 study showed the inverse scaling for midpoint feedback, which was replicated by the current study (on a side note: this replication by itself runs against the title of the paper, i.e., the error-independent effect of sensory uncertainty on motor learning. The within-trial response is also part of motor learning; Kording2004 study was based on this idea as they quantified learning by the within-trial response to perturbations). Other quoted studies (Burge2008, Wei2010, Tsay2021) are the ones that manipulated the endpoint sensory uncertainty, and they indeed showed that sensory uncertainty slows adaptation. 

 Thus, the current study only challenges the three papers claiming an inverse scaling of adaptation and endpoint sensory uncertainty. However, as I put above, the experimental findings here appear intriguing, but they can be accounted for by simple or even trivial explanations that are not adequately discussed in the current manuscript. The initial movement direction, aka the feedforward adaptation, exhibited an abrupt increase in learning following low-uncertainty trials (a 4~5 degree increase). The paper repetitively emphasizes that the effect of sensory uncertainty is independent of movement error. Even with the recognition that explicit aiming strategies might underlie these abrupt changes (and I guess we all agree so), the discussion of relevant findings departs from the current theorization of visuomotor rotation adaptation. For example, the observed initial movement vectors are suggested by the authors to reflect "true adapted state of the motor system". However, it is more or less a consensus in the field that the VMR adaptation to large perturbations consists of implicit and explicit processes. I guess the authors are referring to the former as the "true adapted state." If so, please state it clearly, as readers would not understand what the true adaptation means. Critically, the initial movement vector reflects the sum of implicit and explicit learning; this is the foundation to attribute the abrupt changes following low-uncertainty trials to explicit error correction. The other obvious misread of the literature is on L647 where the authors stated that some studies with no apparent link to strategy actually reported a large drop from adaptation to washout and that this finding thus questioned the validity of initial movement vectors as a reliable estimate of the true level of adaptation. First, I don't understand the reasoning here. Second, the quoted studies are mostly old ones before re-aiming strategies are well understood or measured. Third, the washout trial with an exclusion instruction (i.e., excluding the use of strategy) is widely accepted as a reliable measure of implicit learning. This is just another sign that the authors should interpret and discuss the current findings with a clear view of the current understanding of motor adaptation.

 Going back to the three studies that the current findings appear to contradict. All three of them are probably free from the contamination of explicit strategy. Burge2008 used a relatively small step change of perturbations (8.2 degrees) in both directions with blurring cursor feedback in their experiment 1. These small angles might make the detection of external perturbations hard (see Oh & Schweighoferm 2019), and thus adaptation is mostly driven by slow implicit learning (see their data). Tsay2021 study used task-irrelevant error clamps that presumably elicit implicit adaptation only. Wei2010 study had a zero-mean perturbation size across trials, thus preventing a stable explicit strategy and a slow learning envelope. This is not correctly recognized by the authors (Line 666). Thus, careful examination of the literature and the current findings should lead us to conclude that explicit strategy use overshadows the effect of sensory uncertainty on adaptation, which are primarily implicit in relevant studies.

 In summary, the study did not fulfill its aims through its experimental design since explicit strategy explains away the major findings. The study aims to study the effect of sensory uncertainty when both feedback integration (midpoint feedback if more precisely called) and feedforward adaptation (i.e., endpoint feedback) are present during visuomotor rotation adaptation. However, having midpoint feedback is not equivalent to independently invoking feedback integration, and having endpoint feedback is not equivalent to independently invoking feedforward adaptation. As I put above, midpoint feedback also drives feedforward adaptation, and endpoint feedback can also be part of the feedback process (it just does not lead to within-trial correction). Thus, the theoretical framework of the study and its actual implementation do not align. More importantly, the blurring of cursor feedback, designed to manipulate sensory uncertainty, leads to undesired consequences, e.g., "touching" the ideal direction or the target, which in turn causes strategic corrections. These strategic corrections dominate the otherwise monotonic learning pattern. On the surface, the current study challenges the Bayesian view of sensory uncertainty in motor adaptation; in reality, it is the experimental specifics that produced some intriguing data.  

I suggest the authors correctly quote relevant studies, precisely define their terms, and modestly interpret their findings with the consideration of the current understanding in the field. This means an overhaul of the Introduction and the Discussion, but it would certainly improve the study's contribution to the field. 

 

Minor:

Line147: I have no idea why checking these t-tests would indicate the washout decay is "somewhere between moderate and high uncertainty trials." This statement is wrong on multiple levels. t statistic and its p values are not for directly showing the effect size. If explicit learning is to be quantified, it is a common approach to compute the difference between the last couple of adaptation trials and the first washout trial, under the condition that participants are instructed to refrain from using the re-aiming strategy before the washout. 

Please correct various writing errors, e.g., L809: Th is; Line285, ?; Line475: Experiment 3.

Line288: what caveats?

Line640: combine this small paragraph with the one below. 

Line656: high r square means little for showing the validity of the models. The adaptation pattern here is a simple logarithmic function, and the log(trial) can explain most of the data variance. The additional uncertainty bias terms, retention, and learning betas are icing on the cake. Please show the partial eta squares for each term to drive this message home.

**Have the authors made all data and (if applicable) computational code underlying the findings in their manuscript fully available?**

Reviewer #1: None

Reviewer #2: Yes

PLOS authors have the option to publish the peer review history of their article (what does this mean?). If published, this will include your full peer review and any attached files.

Reviewer #1: No

Reviewer #2: No

Figure Files:

Data Requirements:

Reproducibility:

References:

---

## [Editor Report · Decision Letter 2]

15 Aug 2023

Dear Dr Crossley,

Thank you for your diligent responses to the Reviewer's previous comments and further revisions to the manuscript. We are pleased to inform you that your manuscript 'Error-independent effect of sensory uncertainty on motor learning when both feedforward and feedback control processes are engaged' has been provisionally accepted for publication in PLOS Computational Biology.

Best regards,

Adrian M Haith

Academic Editor

PLOS Computational Biology

Daniele Marinazzo

Section Editor

PLOS Computational Biology

---

## [Editor Report · Acceptance letter]

4 Sep 2023

PCOMPBIOL-D-22-01297R2 

Error-independent effect of sensory uncertainty on motor learning when both feedforward and feedback control processes are engaged

Dear Dr Crossley,

I am pleased to inform you that your manuscript has been formally accepted for publication in PLOS Computational Biology. Your manuscript is now with our production department and you will be notified of the publication date in due course.

With kind regards,

Dorothy Lannert
